# AVOIDING BARREN PLATEAUS VIA GAUSSIAN MIXTURE MODEL

## ABSTRACT

Variational quantum algorithms is one of the most representative algorithms in quantum computing, which has a wide range of applications in quantum machine learning, quantum simulation and other related fields. However, they face challenges associated with the barren plateau phenomenon, especially when dealing with large numbers of qubits, deep circuit layers, or global cost functions, making them often untrainable. In this paper, we propose a novel parameter initialization strategy based on Gaussian Mixture Models. We rigorously prove that, the proposed initialization method consistently avoids the barren plateaus problem for hardware-efficient ansatz with arbitrary length and qubits and any given cost function. Specifically, we find that the gradient norm lower bound provided by the proposed method is independent of the number of qubits $N$ and increases with the circuit depth $L$. Our results strictly highlight the significance of Gaussian Mixture model initialization strategies in determining the trainability of quantum circuits, which provides valuable guidance for future theoretical investigations and practical applications.

## 1 INTRODUCTION

In recent years, the rapid advancement of quantum computing technology has drawn attention to Variational Quantum Algorithms (VQAs) McClean et al. (2016); Cîrstoiu et al. (2020); Cerezo et al. (2022) as a promising quantum algorithm with broad application prospects. In the current era of Noisy Intermediate-Scale Quantum (NISQ) devices Bharti et al. (2022); Arrasmith et al. (2019); Preskill (2018), VQAs provides a feasible approach to solving complex problems, where challenges such as noise and errors in quantum computing devices make large-scale fully quantum computations difficult Benedetti et al. (2019); Jerbi et al. (2023); Cerezo et al. (2021a); Moll et al. (2018). On the other hand, VQAs utilizes Parametrized Quantum Circuits (PQCs), denoted as $V(\boldsymbol{\theta})$, as its quantum computing framework. PQCs serving as a trainable model adjusts its parameters $\boldsymbol{\theta}$ through classical optimization to minimize or maximize a specified cost function. By employing parametrized quantum circuits, VQAs can adapt flexibly to the characteristics of different problems, providing a robust and practical option for quantum computation on NISQ devices Peruzzo et al. (2014); Zhou et al. (2020); Tabares et al. (2023); Pan et al. (2023). VQAs exhibit immense potential across a spectrum of applications, showcasing efficient quantum algorithms that excel in tasks ranging from chemical molecular structure and energy calculations McArdle et al. (2020); Kandala et al. (2017); Hempel et al. (2018) to combinatorial optimization problems Amaro et al. (2022); Akshay et al. (2020) and machine learning Havlíček et al. (2019); Saggio et al. (2021); Schuld et al. (2020); Schuld & Killoran (2019); Zhang et al. (2021); Tian et al. (2023); Zhang et al. (2020); Chen et al. (2020). These applications not only have profound implications for scientific research but also offer innovative solutions for practical applications.

Training VQAs encompasses various methodologies, including gradient-based Sweke et al. (2020); Basheer et al. (2023); Qi et al. (2023) and gradient-free Nelder & Mead (1965); Powell (1964) approaches. However, regardless of the sampling method employed, it is susceptible to encountering the notorious barren plateaus (BP) problem McClean et al. (2018); Arrasmith et al. (2021); Liu et al. (2022); Larocca et al. (2024). The phenomenon of the barren plateau is characterized by the randomized initialization of parameters $\boldsymbol{\theta}$ in VQAs, leading to an exponential vanishing of the cost function gradient along any direction with the increasing number of qubits. We have observed that recent work has employed Lie groups and Lie algebras to provide a unified framework for understanding

the emergence of BP Ragone et al. (2024); Fontana et al. (2024). This framework reveals a close relationship between BP and factors such as noise, the loss function, and circuit structure. Specifically, noise is a significant cause of the barren plateau problem Wang et al. (2021); Stilck França & Garcia-Patron (2021). While depolarizing noise leads to the emergence of BP when the circuit depth $L$ becomes sufficiently large Wang et al. (2021), however, in the case of non-unital noise, barren plateaus do not appear for local cost functions, regardless of the circuit depth Singkanipa & Lidar (2024). The essential cause of the emergence of BP lies in the entanglement within quantum circuits Ortiz Marrero et al. (2021). Numerous strategies have emerged to address this issue, such as optimizing initialization policies Zhang et al. (2022a); Wang et al. (2023); Friedrich & Maziero (2022); Liu et al. (2023), refining circuit structures Liu et al. (2024); Zhao & Gao (2021); Pesah et al. (2021); Cong et al. (2019); Martín et al. (2023); Park & Killoran (2024), or employing local cost functions Arrasmith et al. (2021); Liu et al. (2022). However, whether noise is present or not, avoiding the BP phenomenon for global loss functions remains a challenging problem Cerezo et al. (2021b); Mele et al. (2024). The design of the circuit ansatz is crucial for capturing quantum correlations, including physics-inspired Taube & Bartlett (2006); Wecker et al. (2015); Peruzzo et al. (2014) and hardware-efficient ansatz designs Zhang et al. (2022b). While physics-inspired ansatz exhibits some advantages in certain aspects Wecker et al. (2015); O'Malley et al. (2016), they also face serious challenges in terms of computational resources. On the other hand, hardware-efficient ansatz Kandala et al. (2017) caters to the limitations of NISQ devices, striking a balance between achievability and performance Zhang et al. (2024). At the same time, in this structure, deeper layers exhibit stronger expressibility Ragone et al. (2024); Fontana et al. (2024), and as a result, the BP emerges regardless of the form of the measurement operator or the initial state. The quest for an effective solution to mitigate BP and enhance the versatility of addressing linear combinations in the context of a hardware-efficient ansatz continues to be a forefront challenge in the training of VQAs.

The Gaussian Mixture Model (GMM) Reynolds (2015) is a probability distribution model composed of multiple Gaussian distributions. Each Gaussian distribution, referred to as a component, contributes to the overall mixture distribution. Every component is characterized by its own mean, variance, and weight. This versatile model finds widespread applications in statistics and machine learning Rasmussen (1999); Xuan et al. (2001); Zong et al. (2018), particularly in tasks such as clustering Yang et al. (2012); Manduchi et al. (2021), density estimation Glodek et al. (2013), and generative modeling GM et al. (2020). GMM excels at fitting complex data distributions and, owing to its flexibility and expressive power, is frequently employed for modeling diverse categories of data.

In the training of VQAs, the parameter update expression for the cost function $f(\boldsymbol{\theta})$ based on gradient optimization methods is $f(\boldsymbol{\theta}_{k+1}) = f(\boldsymbol{\theta}_k) - \alpha||\nabla_{\boldsymbol{\theta}} f(\boldsymbol{\theta}_k)||_2^2 + o(\alpha)$, where $\boldsymbol{\theta}_{k+1} = \boldsymbol{\theta}_k - \alpha\nabla_{\boldsymbol{\theta}} f(\boldsymbol{\theta}_k)$, $\alpha$ is the learning rate. Therefore, typically $||\nabla_{\boldsymbol{\theta}} f(\boldsymbol{\theta})||_2^2$ is used to determine whether the cost function $f(\boldsymbol{\theta}) = \text{Tr}[\boldsymbol{O}V(\boldsymbol{\theta})\rho_{in}V^{\dagger}(\boldsymbol{\theta})]$ can be updated. Here, $\boldsymbol{O}$ is an observable, $V(\boldsymbol{\theta})$ is a parameterized quantum circuit, and $\rho_{in}$ is the input quantum state. In this paper, we employ GMM for parameter initialization in VQAs to address the barren plateau problem. Considering arbitrary observables $\boldsymbol{O}$ which can be a single term or a linear combination of terms, by designing specific GMM initialization methods based on $\boldsymbol{O}$, we rigorously prove the following conclusions: (1) When the observable $O$ consists of a single term, the lower bound of $||\nabla_{\boldsymbol{\theta}} f(\boldsymbol{\theta})||_2^2$ is independent of the number of quantum bits $N$ and increases with the circuit length; (2) When $\boldsymbol{O}$ is a linear combination of many terms, the lower bound of $||\nabla_{\boldsymbol{\theta}} f(\boldsymbol{\theta})||_2^2$ increases compared to the single-term case and not decrease; (3) When $\boldsymbol{O}$ consists of non-negative terms, by adjusting GMM, we may achieve a larger lower bound. Therefore, the barren plateau problem does not occur in these scenario, and the model can undergo effective training. This is significant for reducing the cost and saving quantum resources during model training. Additionally, numerical experiments show excellent performance for both local and global cost functions using our method.

## 2 NOTATIONS AND FRAMEWORK

The probability density function of the GMM can be expressed as a weighted sum of individual components. Assuming there are $K$ components, for a given one-dimensional variable $x$, the GMM's probability density function can be represented as:

$$p(x) = \sum_{i=1}^{K} \pi_i \cdot \mathcal{N}(x|\mu_i, \sigma_i^2) \tag{1}$$

where $K$ is the number of Gaussian components, $\pi_i$ is the weight of the ith component, satisfying $\sum_{i=1}^{K} \pi_i = 1$, $\mathcal{N}(x|\mu_i, \sigma_i^2)$ is the probability density function of the ith Gaussian component, with mean $\mu_i$ and variance $\sigma_i^2$. Here are a few rules. Let $\mathcal{G}_0$ be an arbitrary distribution, and if the random variable $\theta$ follows any distribution, it can be expressed as $\theta \sim \mathcal{G}_0$. Furthermore, $\mathcal{G}_1(\sigma^2)$ denotes the Gaussian distribution $\mathcal{N}(0, \sigma^2)$. $\mathcal{G}_2(\sigma^2)$ denotes the first GMM we used, where it's probability density function is $p(x) = \frac{1}{2}\mathcal{N}(x| - \frac{\pi}{2}, \sigma^2) + \frac{1}{2}\mathcal{N}(x|\frac{\pi}{2}, \sigma^2)$. Similarly, $\mathcal{G}_3(\sigma^2)$ is the second GMM, where it's probability density function is $p(x) = \frac{1}{4}\mathcal{N}(x| - \pi, \sigma^2) + \frac{1}{4}\mathcal{N}(x|\pi, \sigma^2) + \frac{1}{2}\mathcal{N}(x|0, \sigma^2)$.

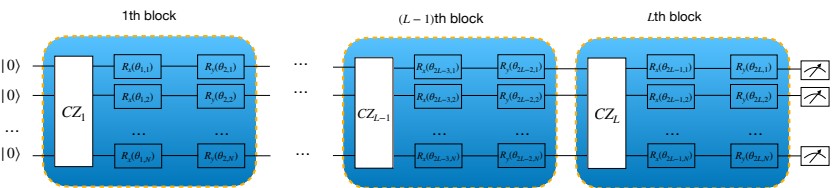

Figure 1: The fundamental framework of the variational quantum circuit, comprising L blocks. Each block begins with the introduction of entanglement through $CZ_l$ gates, followed by the application of $R_x$ and $R_y$ gates on each qubit. Where $CZ_l$ represents any number of CZ gates acting on any two qubits.

In this paper, we employ the ansatz illustrated in Fig.1, which is a typical hardware-efficient ansatz. It involves $N$ qubits and $L$ blocks. Its objective is to minimize the cost function $f(\boldsymbol{\theta}) = \text{Tr}[\boldsymbol{O}V(\boldsymbol{\theta})\rho_{in}V^{\dagger}(\boldsymbol{\theta})]$ by optimizing the parameters $\boldsymbol{\theta}$ within the circuit. In this paper, we assume that $\rho_{in}$ is a pure state. In most cases, $\rho_{in} = |\boldsymbol{0}\rangle\langle\boldsymbol{0}|$ and $|\boldsymbol{0}\rangle = |0\rangle^{\otimes N}$. For an arbitrary observable $\boldsymbol{O} = o_1 \otimes o_2 \otimes ... \otimes o_N$, where $o_i \in \{I, X, Y, Z\}$. We define $I_S := \{n|o_n \neq I, n \in [N]\}$, representing the set of qubits where the observable acts nontrivially, and there are $S$ elements in this set Wang et al. (2023); Zhang et al. (2022a).

For the sake of convenience, let's introduce some notations that will be used in the following theorem. When there are two observables $\boldsymbol{O}_i = o_1^i \otimes o_2^i \otimes ... \otimes o_N^i$ and $\boldsymbol{O}_j = o_1^j \otimes o_2^j \otimes ... \otimes o_N^j$, for all $m \in [N]$, the Pauli matrices at the $m$-th position are denoted by $o_m^i$ and $o_m^j$. We provide the following definitions:

$$S_3^{ij} := |\{m|o_m^i = o_m^j = Z, m \in [N]\}| \tag{2}$$

$$S_{1:3}^{ij} := |\{m|o_m^i = o_m^j \neq I, m \in [N]\}| \tag{3}$$

$$S_{0,3}^{ij} := |\{m|o_m^i = I, o_m^j = Z||o_m^i = Z, o_m^j = I, m \in [N]\}|. \tag{4}$$

We will now delve into the relationship between observables and inactive parameters. Let's assume the observable $\boldsymbol{O}$ is a global observable, i.e., $\boldsymbol{O} = o_1 \otimes o_2 \otimes ... \otimes o_N$, where $\forall k \in \{1, 2, ..., N\}, o_k \in \{X, Y, Z\}$. Let the density matrix of the final quantum state be $\rho_{2L}$, and the quantum state just before the final $R_y$ rotation gate in the last block be $\rho_{2L-1}$. We find that $f(\boldsymbol{\theta}) = \text{Tr}[\boldsymbol{O}\rho_{2L}] = \text{Tr}[\boldsymbol{O}(R_y(\theta_{2L,1}) \otimes R_y(\theta_{2L,2}) \otimes ... \otimes R_y(\theta_{2L,N}))\rho_{2L-1}(R_y^{\dagger}(\theta_{2L,1}) \otimes R_y^{\dagger}(\theta_{2L,2}) \otimes ... \otimes R_y^{\dagger}(\theta_{2L,N}))]$. Then, when $o_k = Y$, we notice that $\forall\theta_{2L,k}, R_y(\theta_{2L,k})YR_y^{\dagger}(\theta_{2L,k}) = Y$. Obviously, in this case, $\theta_{2L,k}$ is independent of the cost function $f(\boldsymbol{\theta})$, making it an "inactive parameter." When the observable $\boldsymbol{O} = Y \otimes Y \otimes ... \otimes Y$, as shown in Fig. 2, all parameters in the last layer of $R_y$ gates are inactive parameters.

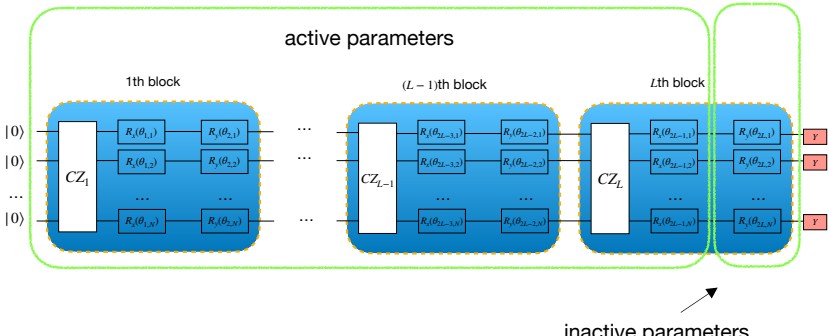

Figure 2: When a term in the observable is $Y$, the parameters in the last block's $R_y(\theta)$ in the ansatz do not contribute to the training. Moreover, when the entire observable consists of $Y$, the $\theta$ parameters in the $R_y$ gates of the last block have no impact on the cost function.

Table 1: On the $i$-th qubit, the parameters in $R_y(\theta)$ and $R_x(\theta)$ are intricately designed, dynamically adjusted based on the distinct Pauli matrices of the observable. When $o_i$ corresponds to $Z$, there are two possible choices for the parameters in $R_x$ and $R_y$.

| The Pauli matrix of $o_i$ | Init method of $R_y(\theta)$ | Init method of $R_x(\theta)$ |
|---|---|---|
| X | $\mathcal{G}_2(\sigma^2)$ | $\mathcal{G}_1(\sigma^2)$ |
| Y | $\mathcal{G}_0$ | $\mathcal{G}_2(\sigma^2)$ |
| Z | $\mathcal{G}_1(\sigma^2)/\mathcal{G}_3(\sigma^2)$ | $\mathcal{G}_1(\sigma^2)/\mathcal{G}_3(\sigma^2)$ |
| I | $\mathcal{G}_0$ | $\mathcal{G}_0$ |

## 3 MAIN RESULTS

We begin by considering the case where the observable consists of only one term, which can be either global or local. Previous research has indicated that avoiding the barren plateau problem for global observables is challenging Sharma et al. (2022); Liu et al. (2022); Cerezo et al. (2021b). Nevertheless, regardless of the specifics, we will rigorously prove that it does not encounter the barren plateau problem when we adopt the GMM as the parameter initialization strategy. The ansatz that we consider is shown in Fig. 1. Here, parameters in different blocks will be initialized using distinct methods, and the initialization approach is determined based on the observable $O$. For convenience, as illustrated in Table 1, we adopt a tabular format to describe the distribution of the parameter $\theta$ in the final block. Now, let's formulate our first theorem.

**Theorem 1** *Consider a VQAs problem defined above, assuming that the parameters $\theta$ in the last block defined in Table 1, and the parameters $\theta$ of the remaining blocks obey the distribution $\mathcal{G}_1(\sigma^2)$, where $\sigma^2 = \frac{1}{2LS}$. Then $\forall q \in \{1, ...2L\}, n \in \{1, ...N\}$, we have*

$$\mathbb{E}_{\boldsymbol{\theta}} \partial_{\theta_{q,n}} f(\boldsymbol{\theta}) = 0 \tag{5}$$

$$\mathbb{E}_{\boldsymbol{\theta}} ||\nabla_{\boldsymbol{\theta}} f(\boldsymbol{\theta})||_2^2 \geq \frac{1}{4} - \frac{1}{8L} \tag{6}$$

*where $\nabla_{\boldsymbol{\theta}} f(\boldsymbol{\theta})$ denotes the gradient of function $f(\boldsymbol{\theta})$ about $\boldsymbol{\theta}$.*

The main idea is outlined here, with the detailed proof provided in Appendix A.2 . First, for different type of parameter distribution, by the relationship among observable, rotation matrix, and CZ operation, we introduce some technical results in Appendix A.1. Then We expand the quantum state $\rho_{\text{out}}$ by the PQCs layer by layer. From the last block, we can prove Eq. (5) based on the lemma in the Appendix A.1. Furthermore, it is easy to see that $\mathbb{E}_{\boldsymbol{\theta}} ||\nabla_{\boldsymbol{\theta}} f(\boldsymbol{\theta})||_2^2$ determines the update of

the cost function $f(\boldsymbol{\theta}) = \text{Tr}[\boldsymbol{O}V(\boldsymbol{\theta})\rho_{in}V^\dagger(\boldsymbol{\theta})]$. And $\mathbb{E}_{\boldsymbol{\theta}} ||\nabla_{\boldsymbol{\theta}} f(\boldsymbol{\theta})||_2^2$ can be expanded into a sum of terms composed of $\text{Tr}^2[\boldsymbol{O}_i\rho_0]$, with coefficients determined by the powers of $\alpha = \mathbb{E}_{\theta \sim \mathcal{G}_1(\sigma^2)} \cos^2 \theta$ and $\beta = \mathbb{E}_{\theta \sim \mathcal{G}_1(\sigma^2)} \sin^2 \theta$. Among these terms, we find that the observable $\boldsymbol{O}_i$ composing with only $I$ or $Z$ that has the largest coefficient. In this case $\text{Tr}^2[\boldsymbol{O}_i\rho_0] = 1$, the lower bound of the gradient norm is then determined by the lower bound of these coefficients, which results in the derivation of Eq. (6) and completes the proof.

The above theorem indicates that, employing our initialization method, the issue of barren plateaus can be consistently avoided, regardless of whether the cost function is global or local. From Eq. (6), it is evident that our norm has a constant lower bound of $\frac{1}{8}$. This is in stark contrast to the exponential lower bound $O\left(\frac{1}{L^N}\right)$ found in previous works for global cost functions Zhang et al. (2022a); Wang et al. (2023). The utilization of GMM significantly improves this lower bound. Additionally, we observe that for certain specific observables, not all parameters $\boldsymbol{\theta}$ in the circuit impact the final value of the cost function $f(\boldsymbol{\theta})$. We refer to those $\boldsymbol{\theta}$ parameters that do not affect the cost function as "inactive parameters", while the others are named "active parameters".

We will now delve into the relationship between observables and inactive parameters. Let's assume the observable $\boldsymbol{O}$ is a global observable, i.e., $\boldsymbol{O} = o_1 \otimes o_2 \otimes ... \otimes o_N$, where $\forall k \in \{1, 2, ..., N\}, o_k \in \{X, Y, Z\}$. Let the density matrix of the final quantum state be $\rho_{2L}$, and the quantum state just before the final $R_y$ rotation gate in the last block be $\rho_{2L-1}$. We find that $f(\boldsymbol{\theta}) = \text{Tr}[\boldsymbol{O}\rho_{2L}] = \text{Tr}[\boldsymbol{O}(R_y(\theta_{2L,1}) \otimes R_y(\theta_{2L,2}) \otimes ... \otimes R_y(\theta_{2L,N}))\rho_{2L-1}(R_y^\dagger(\theta_{2L,1}) \otimes R_y^\dagger(\theta_{2L,2}) \otimes ... \otimes R_y^\dagger(\theta_{2L,N}))])$. Then, when $o_k = Y$, we notice that $\forall \theta_{2L,k}, R_y(\theta_{2L,k})YR_y^\dagger(\theta_{2L,k}) = Y$. Obviously, in this case, $\theta_{2L,k}$ is independent of the cost function $f(\boldsymbol{\theta})$, making it an "inactive parameter." When the observable $\boldsymbol{O} = Y \otimes Y \otimes ... \otimes Y$, as shown in Fig. 2, all parameters in the last layer of $R_y$ gates are inactive parameters.

Using a similar approach, we can also demonstrate that when the cost function is global, for all active parameters $\theta_{q,n}$, $\mathbb{V}ar\partial_{\theta_{q,n}} f(\boldsymbol{\theta}) \geq \frac{1}{8LN}$. This provides an additional perspective on how our method enables escape from the barren plateau.

In Ref. Zhang et al. (2022a), it considered that the observable $\boldsymbol{O}$ contains only a single term. In Ref. Wang et al. (2023), the observable $\boldsymbol{O}$ is extended to a sum of multiple terms, with the cross terms in the gradient norm calculation being non-negative. However, if the coefficients of the terms composing $\boldsymbol{O}$ are negative, these cross terms can become non-positive, complicating the escape from barren plateaus. For example, when $O = O_1 + O_2, \forall q \in \{1, 2, ..., 2L\}, n \in \{1, 2, ..., N\}$, we have $\mathbb{E}_{\boldsymbol{\theta}} \left(\frac{\partial f(\boldsymbol{\theta})}{\partial \theta_{q,n}}\right)^2 = \mathbb{E}_{\boldsymbol{\theta}} \left[\frac{\partial f_1(\boldsymbol{\theta})}{\partial \theta_{q,n}} + \frac{\partial f_2(\boldsymbol{\theta})}{\partial \theta_{q,n}}\right]^2 = \mathbb{E}_{\boldsymbol{\theta}} \left[\frac{\partial f_1(\boldsymbol{\theta})}{\partial \theta_{q,n}}\right]^2 + \mathbb{E}_{\boldsymbol{\theta}} \left[\frac{\partial f_2(\boldsymbol{\theta})}{\partial \theta_{q,n}}\right]^2 + 2 \mathbb{E}_{\boldsymbol{\theta}} \left[\frac{\partial f_1(\boldsymbol{\theta})}{\partial \theta_{q,n}} \frac{\partial f_2(\boldsymbol{\theta})}{\partial \theta_{q,n}}\right]$, where $f_1(\theta) = \text{Tr}(O_1 V(\theta)\rho_{in}V^\dagger(\theta))$ and $f_2(\theta) = \text{Tr}(O_2 V(\theta)\rho_{in}V^\dagger(\theta))$. Ref. Wang et al. (2023) proves that $\mathbb{E}_{\boldsymbol{\theta}} \left[\frac{\partial f_1(\boldsymbol{\theta})}{\partial \theta_{q,n}}\right]^2 + \mathbb{E}_{\boldsymbol{\theta}} \left[\frac{\partial f_2(\boldsymbol{\theta})}{\partial \theta_{q,n}}\right]^2 \geq O\left(\frac{1}{L^S}\right)$, $\mathbb{E}_{\boldsymbol{\theta}} \left[\frac{\partial f_1(\boldsymbol{\theta})}{\partial \theta_{q,n}} \frac{\partial f_2(\boldsymbol{\theta})}{\partial \theta_{q,n}}\right] \geq 0$. However, when $O = O_1 - O_2$, the coefficient in front of the cross term is negative. Therefore, in this case, it cannot be guaranteed that $\mathbb{E}_{\boldsymbol{\theta}} \left(\frac{\partial f(\boldsymbol{\theta})}{\partial \theta_{q,n}}\right)^2 \geq O\left(\frac{1}{L^S}\right)$. But in Theorem 2, we demonstrate that even when $\boldsymbol{O}$ is a linear combination of arbitrary terms, the model remains trainable.

Now let's assume $O = \sum_i O_i - \sum_j O_j$, where $O_i$ and $O_j$ can be either global or local. Also, $\forall O_i, O_j, O_i \neq O_j$. This is the most general form of an observable. Here we randomly select one term from the observable to construct the initialization method. The construction of the last block is detailed in Table 2. Suppose there are $S$ nontrivial Pauli matrices in the selected $\boldsymbol{O}_k$. Additionally, there are $M$ terms that differ from $\boldsymbol{O}_k$ only by replacing Pauli $Z$ with the identity matrix $I$ or vice versa among $\boldsymbol{O}_i$ and $\boldsymbol{O}_j$ at corresponding positions (including the original $\boldsymbol{O}_k$ itself). This setup is because learning a generic Pauli string is challenging, while learning certain subclasses of these strings is easier Nietner (2023). So, if $O$ consists of a single term, then $M = 1$. When $O$ is composed of multiple terms, for example, $O = o_1 + o_2 - o_3 = X \otimes Y \otimes Z \otimes I + Y \otimes Y \otimes Z \otimes I - X \otimes Y \otimes I \otimes Z$, if we choose the first term $o_1$ for initializing $\theta$ according to Table 1, considering that the third term $o_3$ differs from the first term $o_1$ only in the third and fourth Pauli matrices, changing $Z$ to $I$ or $I$ to $Z$, then we have $M = 2$. However, if we choose the second term $o_2$ to initialize $\theta$ according to

Table 1, since the first Pauli matrix of $o_1$ and $o_3$ is $X$ while the first Pauli matrix of $o_2$ is $Y$, neither $o_1$ nor $o_3$ satisfies the condition. Thus $M = 1$ at this time.

As before, the PQC is illustrated in Fig. 1. Now we present our Theorem 2.

Table 2: On the $i$-th qubit, the parameters in $R_y(\theta)$ and $R_x(\theta)$ are intricately designed, dynamically adjusted based on the distinct Pauli matrices of the observable.

| The Pauli matrix of $o_i$ | Init method of $R_y(\theta)$ | Init method of $R_x(\theta)$ |
|---|---|---|
| X | $\mathcal{G}_2(\sigma^2)$ | $\mathcal{G}_1(\sigma^2)$ |
| Y | $\mathcal{G}_1(\sigma^2)$ | $\mathcal{G}_2(\sigma^2)$ |
| Z | $\mathcal{G}_1(\sigma^2)$ | $\mathcal{G}_1(\sigma^2)$ |
| I | $\mathcal{G}_1(\sigma^2)$ | $\mathcal{G}_1(\sigma^2)$ |

**Theorem 2** *Considering the loss function $f(\boldsymbol{\theta}) = Tr\left[\left(\sum_i \boldsymbol{O}_i - \sum_j \boldsymbol{O}_j\right) U(\boldsymbol{\theta})\rho_o U(\boldsymbol{\theta})^\dagger\right]$, where $\boldsymbol{O}_i$ and $\boldsymbol{O}_j$ are arbitrary tensor product forms of Pauli matrices. The parameters in the first $L - 1$ blocks all follow a Gaussian distribution $\mathcal{G}_1(\sigma^2)$, where $\sigma^2 = \frac{1}{2LS}$. Then we randomly select one $O_k$, from either $\boldsymbol{O}_i$ or $\boldsymbol{O}_j$. The parameters in the last block are initialized according to the Pauli matrices in $O_k$ as shown in Table 2. With these considerations, we obtain a lower bound on its squared norm of the gradient:*

$$\mathbb{E}_{\boldsymbol{\theta}} ||\nabla_{\boldsymbol{\theta}} f(\boldsymbol{\theta})||_2^2 \geq M(\frac{1}{4} - \frac{1}{8L}) \tag{7}$$

The full proof are in Appendix A.3. Since the parameter distributions for $Z$ and $I$ are the same here, for $\boldsymbol{O}_k$ itself or by just changing $Z$ to $I$ or $I$ to $Z$ in $\boldsymbol{O}_k$, it can undergo a similar proof using Theorem 1. As for other quadratic terms, they are evidently greater than or equal to 0. For any cross terms, when expanded into a series of summations, it becomes apparent that each term is 0. Therefore, all cross terms are equal to 0. Thus, we obtain Eq. (7) and complete the proof.

From Theorem 2, it can be observed that as the number of terms increases, even if there are some terms with negative coefficients, the lower bound on its norm might become larger. This enables us to update the parameters more effectively. However, when we face a situation where the coefficients in its loss are all non-negative, we propose a new initialization method that can provide a larger lower bound in certain specific cases. Assuming our cost function at this stage is $f(\boldsymbol{\theta}) = Tr\left[\sum_i \boldsymbol{O}_i U(\boldsymbol{\theta})\rho_{\text{in}} U(\boldsymbol{\theta})^\dagger\right]$. Once again, we randomly select a term $\boldsymbol{O}_{k'}$, and following the previous notation, let $S$ denote the number of non-identity matrices in $\boldsymbol{O}_{k'}$. We determine the distribution of $\theta$ in the final layer based on the Pauli matrices in $\boldsymbol{O}_{k'}$, as shown in Table 3. As before, we assume that among the remaining terms, there are $M$ terms that differ from $\boldsymbol{O}_{k'}$ only by replacing Pauli $Z$ with the identity matrix $I$ or vice versa at corresponding positions(including the original $O_{k'}$ itself). We denote the set of indices satisfying these conditions, along with $k'$, as $\mathcal{K}$. Next, we present our Theorem 3.

**Theorem 3** *In accordance with the aforementioned definition of the cost function, the parameters of the $L$-th block in the ansatz are defined as presented in Table 3. The parameters in the preceding $L - 1$ blocks all adhere to a Gaussian distribution $\mathcal{G}_1(\sigma^2)$, where $\sigma^2 = \frac{1}{2LS}$. With these considerations, we derive a lower bound on its norm:*

$$\mathbb{E}_{\boldsymbol{\theta}} ||\nabla_{\boldsymbol{\theta}} f(\theta)||_2^2 \geq M(\frac{1}{4} - \frac{1}{8L}) +$$
$$\sum_{\substack{i,j \in \mathcal{K} \\ i \neq j}} \frac{(2L-1)S_3^{ij}}{2LS}(1 - \frac{1}{2LS})^{2LS_{1:3}^{ij}} e^{-\frac{S_{0,3}^{ij}}{2S}} \tag{8}$$

The proof of this theorem is similar to that of Theorem 2, but there will be differences in the cross terms. The details can be found in Appendix A.4. Theorem 3 informs us that when the objective

Table 3: On the $i$-th qubit, the parameters in $R_y(\theta)$ and $R_x(\theta)$ are intricately designed, dynamically adjusted based on the distinct Pauli matrices of the observable.

| The Pauli matrix of $o_i$ | Init method of $R_y(\theta)$ | Init method of $R_x(\theta)$ |
|:---:|:---:|:---:|
| X | $\mathcal{G}_2(\sigma^2)$ | $\mathcal{G}_1(\sigma^2)$ |
| Y | $\mathcal{G}_1(\sigma^2)$ | $\mathcal{G}_2(\sigma^2)$ |
| Z | $\mathcal{G}_1(\sigma^2)$ | $\mathcal{G}_1(\sigma^2)$ |
| I | $\mathcal{G}_1(\sigma^2)$ | $\mathcal{G}_1(\sigma^2)$ |

function does not contain negative terms, compared to Theorem 2, we can achieve initialization for all parameters using only the distributions $\mathcal{G}_1(\sigma^2)$ and $\mathcal{G}_2(\sigma^2)$, no need for $\mathcal{G}_3(\sigma^2)$. Moreover, in specific cases, the lower bound on its norm is large or equal to the bound proposed in Theorem 2.

## 4 EXPERIMENTS

VQAs play a crucial role in various domains, including the modeling of quantum spins Bharti & Haug (2021), quantum machine learning Romero et al. (2017); Biamonte et al. (2017); Maria Schuld & Petruccione (2015), and quantum chemistry Arute et al. (2020); Levine et al. (2009); Cao et al. (2019). In this section, we embark on a comprehensive exploration of our proposed method, drawing comparisons with existing approaches across the spectrum of local and global cost functions. This comparative analysis aims to illuminate the efficacy and adaptability of our strategy in diverse scenarios, shedding light on its potential to enhance quantum computational tasks in both theoretical modeling and practical applications.

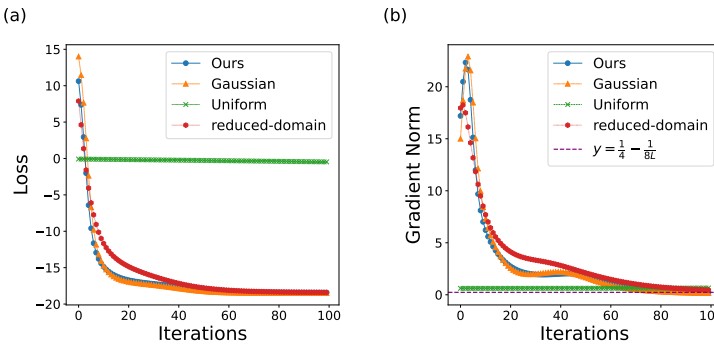

Figure 3: In the training process of the 1D Transverse Field Ising Model, the cost function and gradient norm undergo transformations. Since it is a local cost function, the majority of initialization methods converge to its minimum value.

First, we initially focus on a local observable in the 1D transverse field Ising model (TFIM) Stinchcombe (1973); Heyl et al. (2013), described by the Hamiltonian $H_{\text{TFIM}} = \sum_{i,i+1} Z_i Z_{i+1} - \sum_i X_i$. Setting the initial state $\rho_{in} = |\mathbf{0}\rangle\langle\mathbf{0}|$, with $N = 15$, and $L = 15$, we aim to compute the ground state of the system. We choose the observable $X_1 \otimes I_2 \otimes ... \otimes I_N$ to initialize the circuit parameters. In addition, we compare our proposed method with existing initialization strategies, such as the uniform distribution $\mathcal{U}[-\pi, \pi]$, Gaussian distribution $\mathcal{N}(0, \frac{1}{4S(L+2)})$, and the reduced-domain distribution $\mathcal{U}[-a\pi, a\pi]$, where $a$ is set to 0.07. The experimental results are illustrated in Fig. 3, where (a) depicts the variation of the cost function during the training process, and (b) shows the $\ell_2$ norm of corresponding gradients throughout the optimization. Considering that choosing the observable $Z_1 \otimes Z_2 \otimes ... \otimes I_N$ for initialization could also involve initializing all parameters with a Gaussian distribution, our proposed method offers a broader range of distribution choices. The reduced-domain distribution, similar to the Gaussian distribution, concentrates data around zero. Consequently, our method, along with Gaussian distribution and reduced-domain distribution, proves effective in finding the ground state, significantly outperforming the uniform distribution $\mathcal{U}[-\pi, \pi]$.

(a)                                         (b)

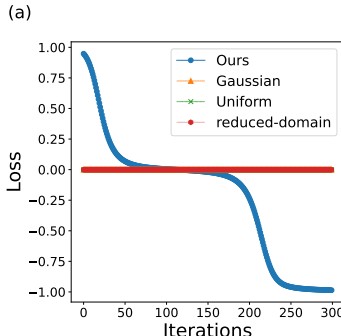 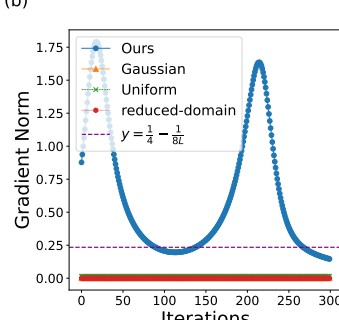

Figure 4: In the training process, when the observable is entirely composed of $X$, the cost function and gradient norm undergo transformations. The gradients for Gaussian, uniform, and reduced-domain distributions remain near zero, resulting in almost non-decreasing cost functions for these distributions. In contrast, our method maintains relatively large gradients throughout the training process and is able to descend to the final results.

Table 4: Comparison of initial gradients norm $\mathbb{E}_{\boldsymbol{\theta}} ||\nabla_{\boldsymbol{\theta}} f(\boldsymbol{\theta})||_2^2$ for different methods at various numbers of qubits.

| Number of qubits $N$ | GMM | Gaussian | Uniform | Reduced-domain |
|---|---|---|---|---|
| 5 | 1.26 | 0.99 | 2.02 | 1.21 |
| 10 | 0.75 | $2.86 \times 10^{-2}$ | 0.41 | $6.22 \times 10^{-2}$ |
| 15 | 0.73 | $1.92 \times 10^{-7}$ | $6.65 \times 10^{-2}$ | $8.56 \times 10^{-4}$ |
| 20 | 0.74 | $3.47 \times 10^{-16}$ | $8.78 \times 10^{-3}$ | $4.61 \times 10^{-6}$ |
| 25 | 0.74 | $2.55 \times 10^{-23}$ | $1.37 \times 10^{-3}$ | $6.87 \times 10^{-8}$ |

However, Gaussian and reduced-domain distributions do not always perform well. For instance, on global cost functions, they can only provide exponential lower bounds, which can not avoid the barren plateau problem in general. Now, we consider the cost function $f(\boldsymbol{\theta}) = \text{Tr}[\boldsymbol{O}U(\boldsymbol{\theta})\rho_{in}U^{\dagger}(\boldsymbol{\theta})]$, where $\boldsymbol{O} = X_1 \otimes X_2 \otimes ... \otimes X_N$, $\rho_{in} = |\boldsymbol{0}\rangle\langle\boldsymbol{0}|$. We set $N = 20$ and $L = 8$, the results are depicted in Fig. 4. Clearly, in this scenario, neither the Gaussian distribution nor the uniform distribution can induce parameter updates, as their gradient norms tend towards zero. In contrast, our method's gradient norm starts with an initial value greater than $\frac{1}{4} - \frac{1}{8L} \approx 0.23$, significantly surpassing others. Moreover, the gradient norm remains within a relatively large range throughout the entire training process. This enables our approach to escape what is commonly referred to as the vanishing gradient problem on plateaus. These observations are entirely consistent with the conclusions drawn in Theorem 1.

Finally, we randomly generate some global observables to calculate their initial gradients. In this case, the cost function is given by $f(\boldsymbol{\theta}) = \text{Tr}[(\sum_{i=1}^{10} \boldsymbol{O}_i - \sum_{j=1}^{10} \boldsymbol{O}_j)U(\boldsymbol{\theta})\rho_{in}U^{\dagger}(\boldsymbol{\theta})]$, where the Pauli matrices in $\boldsymbol{O}_i$ and $\boldsymbol{O}_j$ are randomly selected from $\{X, Y, Z\}$. We set $L$ to be 2 and computed $\mathbb{E}_{\boldsymbol{\theta}} ||\nabla_{\boldsymbol{\theta}} f(\boldsymbol{\theta})||_2^2$ for different numbers of qubits $N$. The results are presented in Table 4. Given that each term is global and excludes Pauli $I$, in this case, $M = 1$. Consequently, according to Theorem 2, our lower bound on $\mathbb{E}_{\boldsymbol{\theta}} ||\nabla_{\boldsymbol{\theta}} f(\boldsymbol{\theta})||_2^2$ is 0.25. From the results, it is evident that with an increase in the number of qubits, the $\mathbb{E}_{\boldsymbol{\theta}} ||\nabla_{\boldsymbol{\theta}} f(\boldsymbol{\theta})||_2^2$ for Gaussian, uniform, and reduced-domain distributions undergoes a sharp reduction. While our method also exhibits a decreasing trend in $\mathbb{E}_{\boldsymbol{\theta}} ||\nabla_{\boldsymbol{\theta}} f(\boldsymbol{\theta})||_2^2$, it aligns closely with the outcome predicted by Theorem 2 and significantly surpasses other methods by several orders of magnitude.

Additionally, we conducted simulation experiments in quantum chemistry to validate the effectiveness of this initialization method. We compared the changes in the loss function as the number of layers $L$ increased, both under noisy and noise-free conditions, as well as the impact of different variances $\sigma^2$ in the GMM on the results. Specific details can be found in the Appendix B.

Table 5: For the $R_y - R_x$ gate structure, we initialize the parameters $\theta$ in both $R_y(\theta)$ and $R_x(\theta)$ gates using a Gaussian distribution $\mathcal{G}_1(\sigma^2)$.

| The Pauli matrix of $o_i$ | Init method of $R_x(\theta)$ | Init method of $R_y(\theta)$ |
|---|---|---|
| X | $\mathcal{G}_2(\sigma^2)$ | $\mathcal{G}_1(\sigma^2)$ |
| Y | $\mathcal{G}_1(\sigma^2)$ | $\mathcal{G}_2(\sigma^2)$ |
| Z | $\mathcal{G}_3(\sigma^2)$ | $\mathcal{G}_3(\sigma^2)$ |
| I | $\mathcal{G}_3(\sigma^2)$ | $\mathcal{G}_3(\sigma^2)$ |

Table 6: For the $R_y - R_x$ gate structure, we initialize the parameters $\theta$ in both $R_y(\theta)$ and $R_x(\theta)$ gates using a Gaussian distribution $\mathcal{G}_1(\sigma^2)$.

| The Pauli matrix of $o_i$ | Init method of $R_x(\theta)$ | Init method of $R_y(\theta)$ |
|---|---|---|
| X | $\mathcal{G}_2(\sigma^2)$ | $\mathcal{G}_1(\sigma^2)$ |
| Y | $\mathcal{G}_1(\sigma^2)$ | $\mathcal{G}_2(\sigma^2)$ |
| Z | $\mathcal{G}_3(\sigma^2)$ | $\mathcal{G}_3(\sigma^2)$ |
| I | $\mathcal{G}_3(\sigma^2)$ | $\mathcal{G}_3(\sigma^2)$ |

## 5 DISCUSSION

We observe that when Pauli matrices are limited to $I$ and $Z$, the CZ gate does not alter their forms. In other words, $CZ^\dagger(o_i \otimes o_j)CZ = o_i \otimes o_j$ for all $o_i, o_j \in \{I, Z\}$. Therefore, $CZ_l$ can be any combination of CZ gates, and it only changes the conditions for 'flip,' which does not affect our results. Also, although our method is specifically effective for the $R_x - R_y$ gate structure, it can be readily extended to other combinations of rotation gates. For instance, as shown in Theorem 2, if we interchange the positions of $R_x$ and $R_y$ in the arrangement of rotation gates, i.e., the arrangement is $R_y - R_x$, then we initialize the parameters of the last block according to Table 6, and the initialization of parameters in other layers follows the distribution $\mathcal{G}_1(\sigma^2)$. Alternatively, when the rotation gates consist of three $R_x - R_y - R_x$ gates, under the same conditions as in Theorem 1, we initialize the parameters of the last block as shown in Table 7, and the initialization of parameters in other layers follows the distribution $\mathcal{G}_1(\sigma^2)$. In both cases, the results are consistent with those of Theorem 1. Certainly, our analysis method remains applicable when using CNOT to provide entanglement.

## 6 CONCLUSION

In this paper, we introduce GMM into the parameter initialization of PQCs to circumvent the notorious barren plateau problem. Results indicate the universality of our approach, as it applies to various cost functions, and we rigorously prove that its gradient norms is no less than $\frac{1}{8}$. We validate our algorithm for diverse problems, which is crucial for VQAs as it enables the training of larger and deeper quantum circuits, unlocking the potential of quantum computation.

While the theorems presented in our paper are tailored to the ansatz in Fig. 1, the applicability of our theorems and proof techniques can extend to other ansatz structures. Furthermore, considering the analogous BP issues in tensor network simulations Liu et al. (2022); Garcia et al. (2023), we anticipate incorporating our method into the initialization of tensor networks in the future. However, due to the sharp-$P$ completeness of classical simulations in tensor networks, even without facing BP, computing their derivatives remains challenging for large-scale problems. In contrast, VQAs can efficiently obtain expected values through quantum devices, making them implementable. Certainly, for effective training of VQAs, overcoming the barren plateau is just one step, as they still face challenges such as local minima Bittel & Kliesch (2021); Anschuetz & Kiani (2022) that need to consider.

We note that recent articles claim all BP-free ansatzes are classically simulable Cerezo et al. (2023). As stated in Ref. Park et al. (2024), HEA can be interpreted as a many-body localized (MBL) system Shtanko et al. (2023), and currently, no efficient classical algorithm can simulate MBL systems for exponentially long times. Additionally, even when using tensor networks to simulate, the barren plateau problem arises when dealing with global loss functionsLiu et al. (2022). Although the work

Table 7: For the $R_x - R_y - R_x$ gate structure, we initialize the parameters $\theta$ in both $R_y(\theta)$ and $R_x(\theta)$ gates using a Gaussian distribution $\mathcal{G}_1(\sigma^2)$.

| $o_i$ | Init method of first $R_x(\theta)$ | Init method of $R_y(\theta)$ | Init method of second $R_x(\theta)$ |
|---|---|---|---|
| X | $\mathcal{G}_3(\sigma^2)$ | $\mathcal{G}_1(\sigma^2)$ | $\mathcal{G}_1(\sigma^2)$ |
| Y | $\mathcal{G}_3(\sigma^2)$ | $\mathcal{G}_2(\sigma^2)$ | $\mathcal{G}_2(\sigma^2)$ |
| Z | $\mathcal{G}_3(\sigma^2)$ | $\mathcal{G}_3(\sigma^2)$ | $\mathcal{G}_3(\sigma^2)$ |
| I | $\mathcal{G}_3(\sigma^2)$ | $\mathcal{G}_3(\sigma^2)$ | $\mathcal{G}_3(\sigma^2)$ |

in Ref. Cerezo et al. (2023) has sparked new thoughts on VQAs, some of its statements require more detailed proof and analysis in future work.

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

# A APPENDIX

## A.1 TECHNICAL LEMMAS

For convenience, let's introduce some notation that will be used in the subsequent proof. Consider a special case where the Pauli matrices $O_i$ and $O_j$ at all corresponding positions are either identical or involve the Pauli Z and the identity matrix. Specifically, $\forall l \in [N]$, the single observables $o_l^i$ and $o_l^j$ at their corresponding positions belong to the set $\{X, X; Y, Y; Z, Z; I, Z; Z, I; I, I\}$. We define:

$$S_1^{ij} := |\{m|o_m^i = o_m^j = X, m \in [N]\}| \tag{9}$$

$$P_0^{ij} := \{m|o_m^i = I || o_m^j = I, m \in [N]\} \tag{10}$$

$$P_{1:3}^{ij} := \{m|o_m^i = o_m^j \neq I, m \in [N]\} \tag{11}$$

$$\tag{12}$$

Also, the random variable $\theta$ is distributed according to $\mathcal{G}_0$, $\mathcal{G}_1(\sigma^2)$, $\mathcal{G}_2(\sigma^2)$, $\mathcal{G}_3(\sigma^2)$, adhering to the same definitions as presented in the main text. Assuming $\theta$ follows the distribution $\mathcal{G}_1(\sigma^2)$, we define $\alpha$, $\beta$, and $\gamma$ as follows:

$$\alpha = \mathop{\mathbb{E}}_{\theta \sim \mathcal{G}_1(\sigma^2)} cos^2\theta = \frac{1 + e^{-2\sigma^2}}{2} \tag{13}$$

$$\beta = \mathop{\mathbb{E}}_{\theta \sim \mathcal{G}_1(\sigma^2)} sin^2\theta = \frac{1 - e^{-2\sigma^2}}{2} \tag{14}$$

$$\gamma = \mathop{\mathbb{E}}_{\theta \sim \mathcal{G}_1(\sigma^2)} cos\theta = e^{-\frac{\sigma^2}{2}} \tag{15}$$

By straightforward application of a Taylor expansion, it is evident that $\alpha \geq 1 - \sigma^2$ and $\beta \geq \sigma^2(1 - \sigma^2)$. Next, we will now present the lemma.

**Lemma 1** *Let $\rho$ be an arbitrary linear operator, $G$ be a Hermitian unitary and $V = e^{-i\frac{\theta}{2}G}$. Consider an arbitrary Hamiltonian operator $O$ that commutes with $G$. Moreover, let $\theta$ be a random variable following an arbitrary distribution, i.e., $\theta \sim \mathcal{G}_0$. Then:*

$$\mathbb{E}_{\theta \sim \mathcal{G}_0} \mathrm{Tr}[OV\rho V^\dagger] = \mathrm{Tr}[O\rho] \tag{16}$$

$$\mathbb{E}_{\theta \sim \mathcal{G}_0} \mathrm{Tr}^2[OV\rho V^\dagger] = \mathrm{Tr}^2[O\rho] \tag{17}$$

$$\mathbb{E}_{\theta \sim \mathcal{G}_0} \frac{\partial}{\partial \theta} \mathrm{Tr}[OV\rho V^\dagger] = 0 \tag{18}$$

*where* $\mathrm{Tr}^2[\cdot] = (\mathrm{Tr}[\cdot])^2$

*Proof.* Consider that $V = e^{-i\frac{\theta}{2}G} = I\cos\left(\frac{\theta}{2}\right) - iG\sin\left(\frac{\theta}{2}\right)$, for any arbitrary operator $O$, we obtain:

$$\mathrm{Tr}[OV\rho V^\dagger] = \mathrm{Tr}\left[O\left(I\cos\left(\frac{\theta}{2}\right) - iG\sin\left(\frac{\theta}{2}\right)\right)\rho\left(I\cos\left(\frac{\theta}{2}\right) + iG\sin\left(\frac{\theta}{2}\right)\right)\right]$$
$$= \frac{1+\cos\theta}{2}\mathrm{Tr}\left[O\rho\right] + \frac{1-\cos\theta}{2}\mathrm{Tr}[OG\rho G] + \frac{\sin\theta}{2}\left(\mathrm{Tr}[iO\rho G] - \mathrm{Tr}[iOG\rho]\right) \tag{19}$$

Given that $G$ is unitary and $[O, G] = 0$, the above expression simplifies to:

$$\mathrm{Tr}[OV\rho V^\dagger] = \mathrm{Tr}[O\rho] \tag{20}$$

Hence, $\mathrm{Tr}[OV\rho V^\dagger]$ is independent of $\theta$. Consequently, for any random variable $\theta$, we establish that $\mathbb{E}_{\theta \sim \mathcal{G}_0} \mathrm{Tr}[OV\rho V^\dagger] = \mathrm{Tr}[O\rho]$, $\mathbb{E}_{\theta \sim \mathcal{G}_0} \mathrm{Tr}^2[OV\rho V^\dagger] = \mathrm{Tr}^2[O\rho]$ and $\mathbb{E}_{\theta \sim \mathcal{G}_0} \frac{\partial}{\partial \theta}\mathrm{Tr}[OV\rho V^\dagger] = 0$.

**Lemma 2** *Let $\rho$ be an arbitrary linear operator, and let $G$ be a Hermitian unitary and $V = e^{-i\frac{\theta}{2}G}$. Consider arbitrary Hamiltonian operator $O_1$, $O_2$, $\widetilde{O}_1$, and $\widetilde{O}_2$, where $O_1$, $O_2$ anti-commute with $G$ and $\widetilde{O}_1$, $\widetilde{O}_2$ commute with $G$, implying $\{O_1, G\} = 0$, $\{O_2, G\} = 0$, $[\widetilde{O}_1, G] = 0$, and $[\widetilde{O}_2, G] = 0$. And $\theta$ is a random variable following a Gaussian distribution $\mathcal{N}(0, \sigma^2)$, i.e., $\theta \sim \mathcal{G}_1(\sigma^2)$. Then:*

$$\mathbb{E}_{\theta \sim \mathcal{G}_1(\sigma^2)} \mathrm{Tr}[O_1 V\rho V^\dagger] = \gamma \mathrm{Tr}[O_1\rho] \tag{21}$$

$$\mathbb{E}_{\theta \sim \mathcal{G}_1(\sigma^2)} \frac{\partial}{\partial \theta}\mathrm{Tr}[O_1 V\rho V^\dagger] = \gamma \mathrm{Tr}[iGO_1\rho] \tag{22}$$

$$\mathbb{E}_{\theta \sim \mathcal{G}_1(\sigma^2)} \mathrm{Tr}[\widetilde{O}_1 V\rho V^\dagger]\mathrm{Tr}[O_1 V\rho V^\dagger] = \gamma \mathrm{Tr}[\widetilde{O}_1\rho]\mathrm{Tr}[O_1\rho] \tag{23}$$

$$\mathbb{E}_{\theta \sim \mathcal{G}_1(\sigma^2)} \frac{\partial}{\partial \theta}\mathrm{Tr}[\widetilde{O}_1 V\rho V^\dagger]\frac{\partial}{\partial \theta}\mathrm{Tr}[O_2 V\rho V^\dagger] = \mathbb{E}_{\theta \sim \mathcal{G}_1(\sigma^2)} \frac{\partial}{\partial \theta}\mathrm{Tr}[\widetilde{O}_1 V\rho V^\dagger]\frac{\partial}{\partial \theta}\mathrm{Tr}[\widetilde{O}_2 V\rho V^\dagger] = 0 \tag{24}$$

$$\mathop{\mathbb{E}}_{\theta\sim\mathcal{G}_1(\sigma^2)} \mathrm{Tr}[O_1 V\rho V^\dagger]\mathrm{Tr}[O_2 V\rho V^\dagger] = \alpha\mathrm{Tr}[O_1\rho]\mathrm{Tr}[O_2\rho] + \beta\mathrm{Tr}[iGO_1\rho]\mathrm{Tr}[iGO_2\rho] \qquad (25)$$

$$\mathop{\mathbb{E}}_{\theta\sim\mathcal{G}_1(\sigma^2)} \frac{\partial}{\partial\theta}\mathrm{Tr}[O_1 V\rho V^\dagger]\frac{\partial}{\partial\theta}\mathrm{Tr}[O_2 V\rho V^\dagger] = \beta\mathrm{Tr}[O_1\rho]\mathrm{Tr}[O_2\rho] + \alpha\mathrm{Tr}[iGO_1\rho]\mathrm{Tr}[iGO_2\rho] \qquad (26)$$

*where $i$ is the imaginary unit.*

*proof.* According to Eq. (19), it can be see that for any operator $O$, we have

$$\mathrm{Tr}[OV\rho V^\dagger] = \frac{1+\cos\theta}{2}\mathrm{Tr}[O\rho] + \frac{1-\cos\theta}{2}\mathrm{Tr}[GOG\rho] + \frac{\sin\theta}{2}(\mathrm{Tr}[iGO\rho] - \mathrm{Tr}[iOG\rho]) \qquad (27)$$

Considering the unitary of $G$ and the conditions $\{O_1, G\} = 0$, as indicated in Eq. (27), we can deduce that

$$\mathrm{Tr}[O_1 V\rho V^\dagger] = \cos\theta\mathrm{Tr}[O_1\rho] + \sin\theta\mathrm{Tr}[iGO_1\rho] \qquad (28)$$

Based on Eq. (28), we obtain that

$$\frac{\partial}{\partial\theta}\mathrm{Tr}[O_1 V\rho V^\dagger] = -\sin\theta\mathrm{Tr}[O_1\rho] + \cos\theta\mathrm{Tr}[iGO_1\rho] \qquad (29)$$

Given that $\mathop{\mathbb{E}}_{\theta\sim\mathcal{G}_1(\sigma^2)}\sin\theta = \mathop{\mathbb{E}}_{\theta\sim\mathcal{G}_1(\sigma^2)}\sin 2\theta = 0$, and combining it with Eq. (20), Eq. (28) and Eq. (29) . Therefore, we can deduce Eq. (21) to Eq. (26).

**Lemma 3** *Let $\rho$, $G$, $V$, $O_1$, $O_2$, $\widetilde{O}_1$ and $\widetilde{O}_2$ be defined in the same manner as presented in Lemma 2. Random variable $\boldsymbol{\theta}$ follows distribution $\mathcal{G}_2(\sigma^2)$. Then*

$$\mathop{\mathbb{E}}_{\theta\sim\mathcal{G}_2(\sigma^2)} \mathrm{Tr}[O_1 V\rho V^\dagger] = 0 \qquad (30)$$

$$\mathop{\mathbb{E}}_{\theta\sim\mathcal{G}_2(\sigma^2)} \frac{\partial}{\partial\theta}\mathrm{Tr}[O_1 V\rho V^\dagger] = 0 \qquad (31)$$

$$\mathop{\mathbb{E}}_{\theta\sim\mathcal{G}_2(\sigma^2)} \mathrm{Tr}[\widetilde{O}_1 V\rho V^\dagger]\mathrm{Tr}[O_1 V\rho V^\dagger] = 0 \qquad (32)$$

$$\mathop{\mathbb{E}}_{\theta\sim\mathcal{G}_2(\sigma^2)} \mathrm{Tr}[\widetilde{O}_1 V\rho V^\dagger]\mathrm{Tr}[\widetilde{O}_2 V\rho V^\dagger] = \mathrm{Tr}[\widetilde{O}_1\rho]\mathrm{Tr}[\widetilde{O}_2\rho] \qquad (33)$$

$$\mathop{\mathbb{E}}_{\theta\sim\mathcal{G}_2(\sigma^2)} \frac{\partial}{\partial\theta}\mathrm{Tr}[\widetilde{O}_1 V\rho V^\dagger]\frac{\partial}{\partial\theta}\mathrm{Tr}[O_2 V\rho V^\dagger] = \mathop{\mathbb{E}}_{\theta\sim\mathcal{G}_2(\sigma^2)} \frac{\partial}{\partial\theta}\mathrm{Tr}[\widetilde{O}_1 V\rho V^\dagger]\frac{\partial}{\partial\theta}\mathrm{Tr}[\widetilde{O}_2 V\rho V^\dagger] = 0 \quad (34)$$

$$\mathop{\mathbb{E}}_{\theta\sim\mathcal{G}_2(\sigma^2)} \mathrm{Tr}[O_1 V\rho V^\dagger]\mathrm{Tr}[O_2 V\rho V^\dagger] = \beta\mathrm{Tr}[O_1\rho]\mathrm{Tr}[O_2\rho] + \alpha\mathrm{Tr}[iGO_1\rho]\mathrm{Tr}[iGO_2\rho] \qquad (35)$$

$$\mathop{\mathbb{E}}_{\theta\sim\mathcal{G}_2(\sigma^2)} \frac{\partial}{\partial\theta}\mathrm{Tr}[O_1 V\rho V^\dagger]\frac{\partial}{\partial\theta}\mathrm{Tr}[O_2 V\rho V^\dagger] = \alpha\mathrm{Tr}[O_1\rho]\mathrm{Tr}[O_2\rho] + \beta\mathrm{Tr}[iGO_1\rho]\mathrm{Tr}[iGO_2\rho] \qquad (36)$$

*proof.* Since $\theta \sim \mathcal{G}_2(\sigma^2)$, we have

$$\underset{\theta\sim\mathcal{G}_2(\sigma^2)}{\mathbb{E}} cos\theta = \frac{1}{2}\int_{-\infty}^{+\infty}\frac{1}{\sqrt{2\pi}\sigma}e^{-\frac{(x+\frac{\pi}{2})^2}{2\sigma^2}}cos(x)dx + \frac{1}{2}\int_{-\infty}^{+\infty}\frac{1}{\sqrt{2\pi}\sigma}e^{-\frac{(x-\frac{\pi}{2})^2}{2\sigma^2}}cos(x)dx \quad (37)$$

$$= -\frac{1}{2}\int_{-\infty}^{+\infty}\frac{1}{\sqrt{2\pi}\sigma}e^{-\frac{x^2}{2\sigma^2}}sin(x)dx + \frac{1}{2}\int_{-\infty}^{+\infty}\frac{1}{\sqrt{2\pi}\sigma}e^{-\frac{x^2}{2\sigma^2}}sin(x)dx \quad (38)$$

$$= 0 \quad (39)$$

By following the similar calculations, we obtain $\underset{\theta\sim\mathcal{G}_2(\sigma^2)}{\mathbb{E}} sin(2\theta) = 0$, $\underset{\theta\sim\mathcal{G}_2(\sigma^2)}{\mathbb{E}} cos^2(\theta) = \beta$, $\underset{\theta\sim\mathcal{G}_2(\sigma^2)}{\mathbb{E}} sin^2(\theta) = \alpha$. Combining them with Eq. (20) and Eq. (28), it is straightforward to have Eq. (30) to Eq. (36).

**Lemma 4** *The definitions of $\rho$, $G$, $V$, $O_1$, $O_2$, $\widetilde{O}_1$ and $\widetilde{O}_2$ align with those outlined in Lemma 2. Random variable $\theta$ follows distribution $\mathcal{G}_3(\sigma^2)$. Then*

$$\underset{\theta\sim\mathcal{G}_3(\sigma^2)}{\mathbb{E}} \text{Tr}[O_1 V\rho V^\dagger] = 0 \quad (40)$$

$$\underset{\theta\sim\mathcal{G}_3(\sigma^2)}{\mathbb{E}} \frac{\partial}{\partial\theta}\text{Tr}[O_1 V\rho V^\dagger] = 0 \quad (41)$$

$$\underset{\theta\sim\mathcal{G}_3(\sigma^2)}{\mathbb{E}} \text{Tr}[\widetilde{O}_1 V\rho V^\dagger]\text{Tr}[O_1 V\rho V^\dagger] = 0 \quad (42)$$

$$\underset{\theta\sim\mathcal{G}_3(\sigma^2)}{\mathbb{E}} \text{Tr}[\widetilde{O}_1 V\rho V^\dagger]\text{Tr}[\widetilde{O}_2 V\rho V^\dagger] = \text{Tr}[\widetilde{O}_1\rho]\text{Tr}[\widetilde{O}_2\rho] \quad (43)$$

$$\underset{\theta\sim\mathcal{G}_3(\sigma^2)}{\mathbb{E}} \frac{\partial}{\partial\theta}\text{Tr}[\widetilde{O}_1 V\rho V^\dagger]\frac{\partial}{\partial\theta}\text{Tr}[O_2 V\rho V^\dagger] = \underset{\theta}{\mathbb{E}} \frac{\partial}{\partial\theta}\text{Tr}[\widetilde{O}_1 V\rho V^\dagger]\frac{\partial}{\partial\theta}\text{Tr}[\widetilde{O}_2 V\rho V^\dagger] = 0 \quad (44)$$

$$\underset{\theta\sim\mathcal{G}_3(\sigma^2)}{\mathbb{E}} \text{Tr}[O_1 V\rho V^\dagger]\text{Tr}[O_2 V\rho V^\dagger] = \alpha\text{Tr}[O_1\rho]\text{Tr}[O_2\rho] + \beta\text{Tr}[iGO_1\rho]\text{Tr}[iGO_2\rho] \quad (45)$$

$$\underset{\theta\sim\mathcal{G}_3(\sigma^2)}{\mathbb{E}} \frac{\partial}{\partial\theta}\text{Tr}[O_1 V\rho V^\dagger]\frac{\partial}{\partial\theta}\text{Tr}[O_2 V\rho V^\dagger] = \beta\text{Tr}[O_1\rho]\text{Tr}[O_2\rho] + \alpha\text{Tr}[iGO_1\rho]\text{Tr}[iGO_2\rho] \quad (46)$$

*proof.* Since $\theta \sim \mathcal{G}_3(\sigma^2)$, we have

$$\underset{\theta\sim\mathcal{G}_3(\sigma^2)}{\mathbb{E}} cos\theta = \frac{1}{4}\int_{-\infty}^{+\infty}\frac{1}{\sqrt{2\pi}\sigma}e^{-\frac{(x+\pi)^2}{2\sigma^2}}cos(x)dx + \frac{1}{4}\int_{-\infty}^{+\infty}\frac{1}{\sqrt{2\pi}\sigma}e^{-\frac{(x-\pi)^2}{2\sigma^2}}cos(x)dx$$

$$+ \frac{1}{2}\int_{-\infty}^{+\infty}\frac{1}{\sqrt{2\pi}\sigma}e^{-\frac{x^2}{2\sigma^2}}cos(x)dx \quad (47)$$

$$= -\frac{1}{4}\int_{-\infty}^{+\infty}\frac{1}{\sqrt{2\pi}\sigma}e^{-\frac{x^2}{2\sigma^2}}cos(x)dx - \frac{1}{4}\int_{-\infty}^{+\infty}\frac{1}{\sqrt{2\pi}\sigma}e^{-\frac{x^2}{2\sigma^2}}cos(x)dx$$

$$+ \frac{1}{2}\int_{-\infty}^{+\infty}\frac{1}{\sqrt{2\pi}\sigma}e^{-\frac{x^2}{2\sigma^2}}cos(x)dx \quad (48)$$

$$= 0 \quad (49)$$

By following the similar calculations, we obtain $\mathbb{E}_{\theta} sin(2\theta) = 0$, $\mathbb{E}_{\theta} cos^2(\theta) = \alpha$, $\mathbb{E}_{\theta} sin^2(\theta) = \beta$. Again using Eq. (20) and Eq. (28), it is straightforward to have Eq. (40) to Eq. (46).

When $O_1 = O_2$ and $\widetilde{O}_1 = \widetilde{O}_2$, we can derive the following corollary:

**Corollary**: *Let $\rho$ be an arbitrary linear operator, and let $G$ be a Hermitian unitary and $V = e^{-i\frac{\theta}{2}G}$. Consider arbitrary quantum observables $O$, where $O$ anti-commute with $G$.*

*If random variable $\theta$ follows distribution $\theta \sim \mathcal{G}_1(\sigma^2)$ or $\theta \sim \mathcal{G}_3(\sigma^2)$ . Then*

$$\mathbb{E}_{\theta} \mathrm{Tr}^2[OV\rho V^\dagger] = \alpha\mathrm{Tr}^2[O\rho] + \beta\mathrm{Tr}^2[iGO\rho], \tag{50}$$

$$\mathbb{E}_{\theta} \left(\frac{\partial}{\partial\theta}\mathrm{Tr}[OV\rho V^\dagger]\right)^2 = \beta\mathrm{Tr}^2[O\rho] + \alpha\mathrm{Tr}^2[iGO\rho]. \tag{51}$$

*If random variable $\theta$ follows a Gaussian mixture model $\theta \sim \mathcal{G}_2(\sigma^2)$. Then*

$$\mathbb{E}_{\theta} \mathrm{Tr}^2[OV\rho V^\dagger] = \beta\mathrm{Tr}^2[O\rho] + \alpha\mathrm{Tr}^2[iGO\rho], \tag{52}$$

$$\mathbb{E}_{\theta} \left(\frac{\partial}{\partial\theta}\mathrm{Tr}[OV\rho V^\dagger]\right)^2 = \alpha\mathrm{Tr}^2[O\rho] + \beta\mathrm{Tr}^2[iGO\rho], \tag{53}$$

For clarity, we employ graphical representations to illustrate the evolution of Pauli matrices. Consider Eq. (45):

$$\mathbb{E}_{\theta\sim\mathcal{G}_3(\sigma^2)} \mathrm{Tr}[O_1V\rho V^\dagger]\mathrm{Tr}[O_2V\rho V^\dagger] = \alpha\mathrm{Tr}[O_1\rho]\mathrm{Tr}[O_2\rho] + \beta\mathrm{Tr}[iGO_1\rho]\mathrm{Tr}[iGO_2\rho] \tag{54}$$

Suppose $O_1 = X, O_2 = Z, G = Y$. Then, $iGO_1 = Z$ and $iGO_2 = -X$. Therefore, $\mathbb{E}_{\theta\sim\mathcal{G}_3(\sigma^2)} \mathrm{Tr}[XV\rho V^\dagger]\mathrm{Tr}[ZV\rho V^\dagger] = \alpha\mathrm{Tr}[X\rho]\mathrm{Tr}[Z\rho] - \beta\mathrm{Tr}[Z\rho]\mathrm{Tr}[X\rho]$. The original operators $O_1$ and $O_2$ are now split into two terms, $X, Z$ and $Z, X$, with coefficients $\alpha$ and $-\beta$ respectively. The corresponding graphical representation, as depicted in Fig. 5, illustrates the evolution of Pauli matrices after applying the gates, with arrows indicating the resulting Pauli matrices and lines representing their parameters.

The following lemma pertains to the transformations of 2-qubit Pauli tensor products after the application of a controlled-Z gate.

**Lemma 5** *Let $CZ$ represent a controlled-Z gate, and $o_i \otimes o_j$ denote a 2-qubit Pauli tensor product, where $o_i$ and $o_j$ are Pauli matrices. When $o_{i'} \otimes o_{j'}$ is equivalent to $CZ^\dagger(o_i \otimes o_j)CZ$, we denote this transformation as $o_i \otimes o_j \to o_{i'} \otimes o_{j'}$. To encapsulate all specific transformations succinctly, we present the following summary:*

$$X \otimes I \leftrightarrow X \otimes Z, X \otimes X \leftrightarrow Y \otimes Y, X \otimes Y \leftrightarrow -Y \otimes X, Y \otimes I \leftrightarrow Y \otimes Z$$

$$Y \otimes Z \leftrightarrow Y \otimes I, Z \otimes I \leftrightarrow Z \otimes I, Z \otimes X \leftrightarrow I \otimes X, Z \otimes Y \leftrightarrow I \otimes Y,$$

$$Z \otimes Z \leftrightarrow Z \otimes Z, I \otimes I \leftrightarrow I \otimes I, I \otimes Z \leftrightarrow I \otimes Z$$

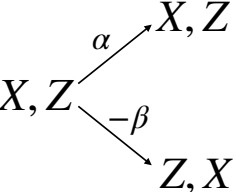

Figure 5: In the scenario where the density matrix $\rho$ remains invariant, the Pauli matrix $XZ$ undergoes a transformation resulting in two components. One component corresponds to $\alpha XZ$, while the other corresponds to $-\beta ZX$.

## A.2 PROOF OF THEOREM 1

Here, we consider an observable with only one term, i.e. $\boldsymbol{O} = o_1 \otimes o_2 \otimes ... \otimes o_N$, where $o_i \in \{I, X, Y, Z\}$. For subsequent calculations, we establish the following notations. We define $\boldsymbol{O_{3:i;1}}$ to mean replacing all the Pauli matrices of $X$ in $O$ with $Z$, and $\boldsymbol{O_{3:i}}$ means replacing all Pauli matrices of $X$ and $Y$ in $O$ with $Z$. The parameterized quantum circuit $U(\boldsymbol{\theta})$ comprising $L$ blocks can be represented as

$$U(\boldsymbol{\theta}) = U_L(\boldsymbol{\theta}_{2L}, \boldsymbol{\theta}_{2L-1})U_{L-1}(\boldsymbol{\theta}_{2L-2}, \boldsymbol{\theta}_{2L-3})...U_1(\boldsymbol{\theta}_2, \boldsymbol{\theta}_1) \tag{55}$$

For each block $U(\boldsymbol{\theta}_l)$, where $l \in \{0, 1, ..., 2L\}$, it can be represented as

$$U_l(\boldsymbol{\theta}_{2l}, \boldsymbol{\theta}_{2l-1}) = R_{2l}(\boldsymbol{\theta}_{2l})R_{2l-1}(\boldsymbol{\theta}_{2l-1})CZ_l \tag{56}$$

where

$$R_{2l}(\boldsymbol{\theta}_{2l}) = e^{-i\frac{\theta_{2l,1}}{2}Y} \otimes e^{-i\frac{\theta_{2l,2}}{2}Y}... \otimes e^{-i\frac{\theta_{2l,N}}{2}Y} \tag{57}$$

$$R_{2l-1}(\boldsymbol{\theta}_{2l-1}) = e^{-i\frac{\theta_{2l-1,1}}{2}X} \otimes e^{-i\frac{\theta_{2l-1,2}}{2}X}... \otimes e^{-i\frac{\theta_{2l-1,N}}{2}X}, \tag{58}$$

$CZ_l$ denotes that the circuit induces entanglement through the inclusion of multiple $CZ$ gates in the $l$-th block.

Next, we consider the intermediate state. For any $k \in \{0, 1, ..., 2L\}$, assuming that the quantum state obtained after passing through the $k$-th block is $\rho_k$, we define

$$\rho_k := \begin{cases} R_k(\boldsymbol{\theta}_k)\rho_{k-1}R_k(\boldsymbol{\theta}_k)^\dagger & \text{for } k = 2l \leq 2L \\ R_k(\boldsymbol{\theta}_k)CZ_{\frac{k+1}{2}}\rho_{k-1}CZ_{\frac{k+1}{2}}^\dagger R_k(\boldsymbol{\theta}_k)^\dagger & \text{for } k = 2l+1 \leq 2L-1 \end{cases} \tag{59}$$

Additionally, we define $I_s := \{m|o_m \neq I, m \in [N]\}$ to denote the set of qubits whose observables act nontrivially. Next, we proceed to prove the content of Theorem 1.

From Theorem 1, we know that when there exists $i \in \{1, 2, ..., N\}$ such that $o_i = I/Z$, the parameters in the last block's $R_x(\boldsymbol{\theta})$ and $R_y(\boldsymbol{\theta})$ gates can follow either the $\mathcal{G}_1(\sigma^2)$ or $\mathcal{G}_3(\sigma^2)$ distribution. For simplicity, we assume the parameters follow the distribution $\mathcal{G}_1(\sigma^2)$, the other case can be proven similarly.

We first consider the case where there is a Pauli matrix $X$ or $Y$ in $O$, i.e., there exists $j$ such that $o_j = X/Y$. Then, the parameters $\theta$ in the last block follow the distributions $\mathcal{G}_1(\sigma^2)$ or $\mathcal{G}_2(\sigma^2)$. According to Eq. (18) and Eq. (31), it is evident that for all $q$ and $n$, $\mathbb{E}_{\theta} \partial_{\theta_{q,n}} f(\boldsymbol{\theta}) = 0$.

Furthermore, when all the Pauli matrices in $O$ are either $I$ or $Z$, we proceed as follows. Assume $\theta_{q,n}$ is in the last block, i.e., $q = 2L - 1$ or $q = 2L$. If the $n$-th Pauli matrix of $O$ is $o_n = I$, then according to Eq. (16) and Eq. (18), it is easy to see that $\mathbb{E}_{\theta} \partial_{\theta_{q,n}} f(\boldsymbol{\theta}) = 0$.

When $o_n = Z$, using Eq. (21) and Eq. (22), the Pauli matrix inevitably transforms into $X$ or $Y$. Combining this with Eq. (21) and Lemma 5, in the final $\text{Tr}[O'\rho]$, the Pauli matrix at the $n$-th position of $O'$ must be $X$ or $Y$. Furthermore, due to $\langle 0|X|0 \rangle = \langle 1|X|1 \rangle = \langle 0|Y|0 \rangle = \langle 1|Y|1 \rangle = 0$, we have $\mathbb{E}_{\theta} \partial_{\theta_{q,n}} f(\boldsymbol{\theta}) = 0$.

When $q \in \{1, ..., 2L - 2\}$, we have:

$$\mathbb{E}_{\boldsymbol{\theta}} \partial_{\theta_{q,n}} f(\boldsymbol{\theta}) = \mathbb{E}_{\boldsymbol{\theta_1}} ... \mathbb{E}_{\boldsymbol{\theta_{2L}}} \partial_{\theta_{q,n}} \text{Tr}[\boldsymbol{O}\rho_{2L}] \tag{60}$$

$$= \gamma^{S_3} \mathbb{E}_{\boldsymbol{\theta_1}} ... \mathbb{E}_{\boldsymbol{\theta_{2L-1}}} \partial_{\theta_{q,n}} \text{Tr}[\boldsymbol{O}\rho_{2L-1}] \tag{61}$$

$$= \gamma^{2S_3} \mathbb{E}_{\boldsymbol{\theta_1}} ... \mathbb{E}_{\boldsymbol{\theta_{2L-2}}} \partial_{\theta_{q,n}} \text{Tr}[CZ_l^{\dagger} \boldsymbol{O} CZ_l \rho_{2L-2}] \tag{62}$$

$$= \gamma^{2S_3} \mathbb{E}_{\boldsymbol{\theta_1}} ... \mathbb{E}_{\boldsymbol{\theta_{2L-2}}} \partial_{\theta_{q,n}} \text{Tr}[\boldsymbol{O}\rho_{2L-2}] \tag{63}$$

$$= \gamma^{(2L-q-1)S_3} \mathbb{E}_{\boldsymbol{\theta_1}} ... \mathbb{E}_{\boldsymbol{\theta_q}} \partial_{\theta_{q,n}} \text{Tr}[\boldsymbol{O}\rho_q] \tag{64}$$

According to Eq. (16) and (21), we can infer that when $n \in I_s$, the expectation of $\theta_n$ yields $\gamma$, and when $n \notin I_s$, the expectation of $\theta_n$ results in a constant 1. Thus, we obtain Eq. (61). Similarly, we can derive Eq. (62). Eq. (63) is derived from Lemma 5. By repeating this process, we arrive at Eq. (64).

We are currently directing our attention to the subscript $n$. If $n \notin I_S$, then, based on Eq. (18), we can obtain,

$$\mathbb{E}_{\boldsymbol{\theta_1}} ... \mathbb{E}_{\boldsymbol{\theta_q}} \partial_{\theta_{q,n}} \text{Tr}[O\rho_q] = 0 \tag{65}$$

which means

$$\mathbb{E}_{\boldsymbol{\theta_1}} ... \mathbb{E}_{\boldsymbol{\theta_{2L}}} \partial_{\theta_{q,n}} f(\boldsymbol{\theta}) = 0. \tag{66}$$

When $n \in I_S$, according to Eq. (22), we have

$$\mathbb{E}_{\boldsymbol{\theta_1}} ... \mathbb{E}_{\boldsymbol{\theta_q}} \partial_{\theta_{q,n}} \text{Tr}[O\rho_{2L-2}] = \gamma^{S_3} \mathbb{E}_{\boldsymbol{\theta_1}} ... \mathbb{E}_{\boldsymbol{\theta_{q-1}}} \partial_{\theta_{q,n}} \text{Tr}[O'\rho_{2L-2}] \tag{67}$$

Among these, $O'$ entails transforming the Pauli $Z$ matrix at the $n$th position of the Hamiltonian $O$ into $Y$ or $-X$. Subsequently, Eq. (16) and Eq. (21) elucidate that applying an expectation to $\boldsymbol{\theta_1}, \boldsymbol{\theta_2}, ..., \boldsymbol{\theta_q}$ does not alter the form of the observable but merely augments certain coefficients from the previous state. Additionally, considering that the observable at this juncture comprises only $Y$ or $-X$ at the $n$th position, with the remaining positions being $Z$ or $I$, Lemma 5 implies that we have

$$\mathbb{E}_{\boldsymbol{\theta_1}} ... \mathbb{E}_{\boldsymbol{\theta_{2L}}} \partial_{\theta_{q,n}} \text{Tr}[O\rho_{2L-2}] = \gamma^c \text{Tr}[O''\rho_0] \tag{68}$$

Here, $c$ is a constant greater than or equal to $L$ and less than or equal to $2L$. Considering that the observable $O''$ involves the Pauli operators $X$ or $Y$ at position $n$, and $\langle 0|X|0 \rangle = 0, \langle 0|Y|0 \rangle = 0$, we obtain

$$\mathbb{E}_{\boldsymbol{\theta_1}} \cdots \mathbb{E}_{\boldsymbol{\theta_{2L}}} \partial_{\theta_{q,n}} \mathrm{Tr}[O \rho_{2L-2}] = 0 \tag{69}$$

Thus far, we have successfully demonstrated that its expectation is equal to 0. Thus we complete the proof of Eq. (5).

Next, we will establish the lower bound of its gardient norm. That is we prove Eq. (6) in Theorem 1. Note that

$$\mathbb{E}_{\boldsymbol{\theta}} ||\nabla_{\boldsymbol{\theta}} f(\boldsymbol{\theta})||^2 = \sum_{q=1}^{2L} \sum_{n=1}^{N} \mathbb{E}_{\boldsymbol{\theta}} \left( \frac{\partial f(\boldsymbol{\theta})}{\partial \theta_{q,n}} \right)^2$$

$$= \sum_{q=1}^{2L} \sum_{n \in I_S} \mathbb{E}_{\boldsymbol{\theta}} \left( \frac{\partial f(\boldsymbol{\theta})}{\partial \theta_{q,n}} \right)^2 + \sum_{q=1}^{2L} \sum_{n \notin I_S} \mathbb{E}_{\boldsymbol{\theta}} \left( \frac{\partial f(\boldsymbol{\theta})}{\partial \theta_{q,n}} \right)^2$$

$$\geq \sum_{q=1}^{2L} \sum_{n \in I_S} \mathbb{E}_{\boldsymbol{\theta}} \left( \frac{\partial f(\boldsymbol{\theta})}{\partial \theta_{q,n}} \right)^2 \tag{70}$$

For each term within the first $L-1$ blocks of $\mathbb{E}_{\boldsymbol{\theta}}(\frac{\partial f(\boldsymbol{\theta})}{\partial \theta_{q,n}})^2$, it follows that

$$\mathbb{E}_{\boldsymbol{\theta}} \left( \frac{\partial f(\boldsymbol{\theta})}{\partial \theta_{q,n}} \right)^2 = \mathbb{E}_{\boldsymbol{\theta}} \left( \frac{\partial}{\partial \theta_{q,n}} \mathrm{Tr}[\boldsymbol{O} \rho_{2L}] \right)^2 \tag{71}$$

$$= \mathbb{E}_{\boldsymbol{\theta_1}} \cdots \mathbb{E}_{\boldsymbol{\theta_{2L}}} \left( \frac{\partial}{\partial \theta_{q,n}} \mathrm{Tr}[\boldsymbol{O} R_{2L}(\boldsymbol{\theta_{2L}}) \rho_{2L-1} R_{2L}^\dagger(\boldsymbol{\theta_{2L}})] \right)^2 \tag{72}$$

$$\geq \alpha^{S_1+S_3} \mathbb{E}_{\boldsymbol{\theta_1}} \cdots \mathbb{E}_{\boldsymbol{\theta_{2L-1}}} \left( \frac{\partial}{\partial \theta_{q,n}} \mathrm{Tr}[\boldsymbol{O}_{3:i;1} \rho_{2L-1}] \right)^2 \tag{73}$$

$$= \alpha^{S_1+S_3} \mathbb{E}_{\boldsymbol{\theta_1}} \cdots \mathbb{E}_{\boldsymbol{\theta_{2L-1}}} \left( \frac{\partial}{\partial \theta_{q,n}} \mathrm{Tr}[\boldsymbol{O}_{3:i;1} R_{2L-1}(\boldsymbol{\theta_{2L-1}}) CZ_L \rho_{2L-2} CZ_L^\dagger R_{2L}^\dagger(\boldsymbol{\theta_{2L-1}}) \right)^2 \tag{74}$$

$$\geq \alpha^{S_1+S_3} \alpha^{S_1+S_3+S_2} \mathbb{E}_{\boldsymbol{\theta_1}} \cdots \mathbb{E}_{\boldsymbol{\theta_{2L-2}}} \left( \frac{\partial}{\partial \theta_{q,n}} \mathrm{Tr}[\boldsymbol{O}_{3:i} CZ_L \rho_{2L-2} CZ_L^\dagger] \right)^2 \tag{75}$$

$$= \alpha^{S_1+S_3} \alpha^{S_1+S_3+S_2} \mathbb{E}_{\boldsymbol{\theta_1}} \cdots \mathbb{E}_{\boldsymbol{\theta_{2L-2}}} \left( \frac{\partial}{\partial \theta_{q,n}} \mathrm{Tr}[\boldsymbol{O}_{3:i} \rho_{2L-2}] \right)^2 \tag{76}$$

$$\geq \alpha^{S_1+S_3} \alpha^{S(2L-1-q)} \mathbb{E}_{\boldsymbol{\theta_1}} \cdots \mathbb{E}_{\boldsymbol{\theta_q}} \left( \frac{\partial}{\partial \theta_{q,n}} \mathrm{Tr}[\boldsymbol{O}_{3:i} \rho_q] \right)^2 \tag{77}$$

In Eq. (73), the formulation arises from the utilization of Eq. (50) when $n$ is in $I_{s_3}$ and Eq. (52) when $n$ is in $I_{s_1}$, contributing a parameter $\alpha$ for each term. Conversely, when $n$ is in either $I_{s_0}$ or $I_{s_2}$, Eq. (17) is employed without altering the preceding coefficients. Through analogous analysis, Eq. (75) is derived. Eq. (76) is a consequence of the deductions stemming from Lemma 5. By iterating through these steps, we arrive at Eq. (77).

$$\mathbb{E}_{\boldsymbol{\theta}}(\frac{\partial f(\boldsymbol{\theta})}{\partial \theta_{q,n}})^2 \geq \alpha^{S_1+S_3}\alpha^{S(2L-1-q)}\alpha^{S-1}\beta \mathbb{E}_{\boldsymbol{\theta_1}} \cdots \mathbb{E}_{\boldsymbol{\theta_{q-1}}} (\mathrm{Tr}[\boldsymbol{O}_{3:i}\rho_{q-1}])^2 \tag{78}$$

$$\geq \alpha^{S_1+S_3}\alpha^{S(2L-1-q)}\alpha^{S-1}\beta\alpha^{S(q-1)}\mathrm{Tr}^2[\boldsymbol{O}_{3:i}\rho_0] \tag{79}$$

$$\geq \alpha^{2LS-1}\beta \tag{80}$$

$$\geq (1-\sigma^2)^{2LS-1}\sigma^2(1-\sigma^2) \tag{81}$$

$$= \frac{1}{2LS}(1-\frac{1}{2LS})^{2LS} \tag{82}$$

$$\geq \frac{1}{8LS} \tag{83}$$

In Eq. (78), the coefficient $\beta$ is determined by taking the expectation with respect to $\theta_{q,n}$ based on Eq. (51). Here, we retain the terms with the coefficient $\beta$ instead of $\alpha$. The remaining $\alpha^{S-1}$ terms remain consistent with Eq. (50). Eq. (79) follows a process similar to Eq. (77), obtained by taking the expectation over the remaining $\theta$. Considering $\mathrm{Tr}[\boldsymbol{O_{3:i}\rho_0}] = 1$, $S_1 + S_3 \leq S$, and $\alpha < 1$, we arrive at Eq. (80). Eq. (81) is derived from a Taylor expansion. Taking into account $h(x) = (1-\frac{1}{x})^x$ being monotonically increasing when $x \geq 2$, Eq. (83) is thus proven.

Applying the identical methodology for analysis, we can similarly derive the same results for the $R_X$ rotation layer in the final block. Thus, we can conclude that

$$\mathbb{E}_{\boldsymbol{\theta}}||\nabla_{\boldsymbol{\theta}}f(\boldsymbol{\theta})||^2 \geq \sum_{q=1}^{2L-1}\sum_{n\in I_S}\mathbb{E}_{\boldsymbol{\theta}}(\frac{\partial f(\boldsymbol{\theta})}{\partial \theta_{q,n}})^2$$

$$\geq \sum_{q=1}^{2L-1}\sum_{n\in I_S}\frac{1}{8LS}$$

$$= (2L-1) \times S \times \frac{1}{8LS}$$

$$= \frac{1}{4} - \frac{1}{8L} \tag{84}$$

### A.3 Proof of Theorem 2

Before proving Theorem 2, let's first consider a special case where both $O_i$ and $O_j$ are global. We can provide the following lemma:

**Lemma 6** *Considering a quantum circuit $U(\boldsymbol{\theta})$ with $N$ qubits, initialized with $\rho_0$ as a pure state, and employing a hardware-efficient ansatz with $L$ blocks, as depicted in Fig. 1, the cost function is defined as $f(\boldsymbol{\theta}) = Tr[(\sum_i \boldsymbol{O}_i - \sum_j \boldsymbol{O}_j)U(\boldsymbol{\theta})\rho_o U(\boldsymbol{\theta})^\dagger]$, where observable $\boldsymbol{O}_i, \boldsymbol{O}_j$ are global observables, denoted as $o_i, o_j \in \{X, Y, Z\}$. Randomly choose either $\boldsymbol{O}_i$ or $\boldsymbol{O}_j$ and initialize it in accordance with the procedure outlined in Theorem 2. Consequently, we obtain:*

$$\mathbb{E}_{\theta}||\nabla_{\boldsymbol{\theta}}f(\boldsymbol{\theta})||_2^2 \geq \frac{1}{4} - \frac{1}{8L} \tag{85}$$

*proof:* Without loss of generality, let us opt to specify $\boldsymbol{O}_1$ and initialize the parameters within $U(\boldsymbol{\theta})$ following the methodology expounded in Theorem 1. Subsequently, we have

$$\mathbb{E}_{\boldsymbol{\theta}} ||\nabla_{\boldsymbol{\theta}} f(\boldsymbol{\theta})||^2 = \sum_{q,n} \mathbb{E}_{\boldsymbol{\theta}} \left( \frac{\partial f(\boldsymbol{\theta})}{\partial \theta_{q,n}} \right)^2 \tag{86}$$

$$= \sum_{q,n} \mathbb{E}_{\boldsymbol{\theta}} \left( \sum_i \frac{\partial f_i(\boldsymbol{\theta})}{\partial \theta_{q,n}} - \sum_j \frac{\partial f_j(\boldsymbol{\theta})}{\partial \theta_{q,n}} \right)^2 \tag{87}$$

$$= \sum_{q,n} \mathbb{E}_{\boldsymbol{\theta}} \left( \sum_i \frac{\partial f_i(\boldsymbol{\theta})}{\partial \theta_{q,n}} \right)^2 - 2 \sum_{q,n} \mathbb{E}_{\boldsymbol{\theta}} \left( \sum_{i,j} \frac{\partial f_i(\boldsymbol{\theta})}{\partial \theta_{q,n}} \cdot \frac{\partial f_j(\boldsymbol{\theta})}{\partial \theta_{q,n}} \right) + \sum_{q,n} \mathbb{E}_{\boldsymbol{\theta}} \left( \sum_j \frac{\partial f_j(\boldsymbol{\theta})}{\partial \theta_{q,n}} \right)^2 \tag{88}$$

$$= \sum_{q,n,i} \mathbb{E}_{\boldsymbol{\theta}} \left( \frac{\partial f_i(\boldsymbol{\theta})}{\partial \theta_{q,n}} \right)^2 + \sum_{q,n,i_1 \neq i_2} \mathbb{E}_{\boldsymbol{\theta}} \left( \frac{\partial f_{i_1}(\boldsymbol{\theta})}{\partial \theta_{q,n}} \cdot \frac{\partial f_{i_2}(\boldsymbol{\theta})}{\partial \theta_{q,n}} \right)$$

$$- 2 \sum_{q,n,i,j} \mathbb{E}_{\boldsymbol{\theta}} \left( \frac{\partial f_i(\boldsymbol{\theta})}{\partial \theta_{q,n}} \cdot \frac{\partial f_j(\boldsymbol{\theta})}{\partial \theta_{q,n}} \right)$$

$$+ \sum_{q,n,j} \mathbb{E}_{\boldsymbol{\theta}} \left( \frac{\partial f_j(\boldsymbol{\theta})}{\partial \theta_{q,n}} \right)^2 + \sum_{q,n,j_1 \neq j_2} \mathbb{E}_{\boldsymbol{\theta}} \left( \frac{\partial f_{j_1}(\boldsymbol{\theta})}{\partial \theta_{q,n}} \cdot \frac{\partial f_{j_2}(\boldsymbol{\theta})}{\partial \theta_{q,n}} \right) \tag{89}$$

We expand the function $f(\boldsymbol{\theta})$, resulting in Eq. (89). Here, $f_i(\boldsymbol{\theta}) = \text{Tr}[\boldsymbol{O}_i U(\boldsymbol{\theta}) \rho_0 U(\boldsymbol{\theta})^\dagger]$ and $f_j(\boldsymbol{\theta}) = \text{Tr}[\boldsymbol{O}_j U(\boldsymbol{\theta}) \rho_0 U(\boldsymbol{\theta})^\dagger]$. Moving forward, let's consider the cross terms. Without loss of generality, let's examine each element in the third term. Let's denote $O_i = \vec{\sigma}_{i,2L} = \sigma_{1,i,2L} \otimes \sigma_{2,i,2L} \otimes ... \otimes \sigma_{N,i,2L}$ and $O_j = \vec{\sigma}_{j,2L} = \widetilde{\sigma}_{1,j,2L} \otimes \widetilde{\sigma}_{2,j,2L} \otimes ... \otimes \widetilde{\sigma}_{N,j,2L}$. Next, we focus on the evolution of these Pauli matrices throughout the process, we have:

$$\mathbb{E}_{\boldsymbol{\theta}} \left( \frac{\partial f_i(\boldsymbol{\theta})}{\partial \theta_{q,n}} \frac{\partial f_j(\boldsymbol{\theta})}{\partial \theta_{q,n}} \right) = \mathbb{E}_{\boldsymbol{\theta}} \left( \frac{\partial}{\partial \theta_{q,n}} \text{Tr}[\vec{\sigma}_{i,2L} \rho_{2L}] \frac{\partial}{\partial \theta_{q,n}} \text{Tr}[\vec{\sigma}_{j,2L} \rho_{2L}] \right) \tag{90}$$

$$= \mathbb{E}_{\boldsymbol{\theta}} \left( \text{Tr}[\vec{\sigma}_{i,2L} R_{2L}(\boldsymbol{\theta}) \frac{\partial \rho_{2L-1}}{\partial \theta_{q,n}} R_{2L}^\dagger(\boldsymbol{\theta})] \text{Tr}[\vec{\sigma}_{j,2L} R_{2L}(\boldsymbol{\theta}) \frac{\partial \rho_{2L-1}}{\partial \theta_{q,n}} R_{2L}^\dagger(\boldsymbol{\theta})] \right) \tag{91}$$

$$= \sum_{k_1} h_{k_1} \mathbb{E}_{\boldsymbol{\theta}} \left( \text{Tr}[\vec{\sigma}_{i,2L-1}^{k_1} \frac{\partial \rho_{2L-1}}{\partial \theta_{q,n}}] \text{Tr}[\vec{\sigma}_{j,2L-1}^{k_1} \frac{\partial \rho_{2L-1}}{\partial \theta_{q,n}}] \right) \tag{92}$$

$$= \sum_{k_2} h_{k_2} \mathbb{E}_{\boldsymbol{\theta}} \left( \text{Tr}[CZ^\dagger \vec{\sigma}_{i,2L-2}^{k_2} CZ \frac{\partial \rho_{2L-2}}{\partial \theta_{q,n}}] \text{Tr}[CZ^\dagger \vec{\sigma}_{j,2L-2}^{k_2} CZ \frac{\partial \rho_{2L-2}}{\partial \theta_{q,n}}] \right) \tag{93}$$

$$= \sum_{k_2'} h_{k_2'} \mathbb{E}_{\boldsymbol{\theta}} \left( \text{Tr}[\vec{\sigma}_{i,2L-2}^{k_2'} \frac{\partial \rho_{2L-2}}{\partial \theta_{q,n}}] \text{Tr}[\vec{\sigma}_{j,2L-2}^{k_2'} \frac{\partial \rho_{2L-2}}{\partial \theta_{q,n}}] \right) \tag{94}$$

$$\cdots \tag{95}$$

$$= \sum_{k_{2L}'} h_{k_{2L}'} \text{Tr}[\vec{\sigma}_{i,0}^{k_{2L}'} \rho_0] \text{Tr}[\vec{\sigma}_{j,0}^{k_{2L}'} \rho_0] \tag{96}$$

Among these, the coefficients $h_{k_1}, h_{k_2}, h_{k_2'}, \ldots, h_{k_{2L}}, h_{k_{2L}'}$ take the form $\pm \alpha^{g_1} \beta^{g_2} \gamma^{g_3}$, where $g_1, g_2, g_3 \in \mathbb{N}$. $\vec{\sigma}_{i,0}^{k_{2L}'}, \vec{\sigma}_{i,0}^{k_{2L}}, \ldots, \vec{\sigma}_{i,2L-1}^{k_1}, \vec{\sigma}_{i,2L}, \vec{\sigma}_{j,0}^{k_{2L}'}, \vec{\sigma}_{j,0}^{k_{2L}}, \ldots, \vec{\sigma}_{j,2L-1}^{k_1}, \vec{\sigma}_{j,2L}$ are all in the form of Pauli matrix tensor product. Furthermore, since $\boldsymbol{O_i}$ and $\boldsymbol{O_j}$ are both globally observable operators, and $\boldsymbol{O_i} \neq \boldsymbol{O_j}$, there exists $k \in [N]$ such that the Pauli matrix on the $k$-th qubit of $\sigma_{k,i,2L}$ and $\widetilde{\sigma}_{k,j,2L}$ is one of the cases $\{X,Y;Y,X;X,Z;Z,X;Y,Z;Z,Y\}$. Next, we will prove that for all these combinations, $\mathbb{E}_{\boldsymbol{\theta}} \left( \frac{\partial f_i(\boldsymbol{\theta})}{\partial \theta_{q,n}} \frac{\partial f_j(\boldsymbol{\theta})}{\partial \theta_{q,n}} \right) = 0$. Without loss of generality, let's assume that there exists $k$ such that the $k$-th position of $\sigma_{k,i,2L}$ is $X$ and the $k$-th position of $\widetilde{\sigma}_{k,j,2L}$ is $Z$.

Next, let's consider the changes in observables. According to Lemma 1, 2, 3, and 4, after the last block's $R_y$ rotation gate, regardless of the distribution followed by $\theta$ in $R_x(\theta)$, based on Eq. (25), Eq. (35) and Eq. (45), the value at position $k$ becomes $\{X, Z\}$ or $\{Z, X\}$, the coefficients for the other terms are zero. However, different distributions will result in varying coefficients in front of $\{X, Z\}$ or $\{Z, X\}$. $\{X, Z\}, \{Z, X\}$ remains $\{X, Z\}, \{Z, X\}$ or 0 after the $R_x$ rotation gate, according to Eq. (23), Eq. (33) and Eq. (43). If it's non-zero, according to Lemma 5, the $CZ$ operation can transform the original $X$ or $Y$ into $X$ or $Y$, without changing them into $Z$ or $I$. Similarly, it cannot transform $Z$ and $I$ into $X$ or $Y$. If, after the application of $CZ$, the original Pauli matrix undergoes a change, such as turning $X$ into $Y$ or $Z$ into $I$, we refer to this process as a "flip." Clearly, for any observable $C = c_1 \otimes c_2 \otimes ... \otimes c_n$, if it aims to achieve a "flip" operation at its $k$-th position, it must satisfy the condition that the Pauli matrix at the $(k-1)$-th position belongs to $X, Y$, the Pauli matrix at the $(k+1)$-th position belongs to $I, Z$, or the Pauli matrix at the $(k-1)$-th position belongs to $I, Z$, and the Pauli matrix at the $(k+1)$-th position belongs to $Z, I$. Therefore, after the CZ entanglement gate, its situation becomes one of $\{X, Z; Z, X; Y, Z; Z, Y; X, I; I, X; Y, I; I, Y\}$. Furthermore, taking partial derivatives with respect to any position $\theta_{q,n}$ only alters the coefficients in front, and it does not lead to the appearance of the four possible combinations $\{I, I; Z, Z; I, Z; Z, I\}$ for Pauli matrices.

This analysis applies to each block similarly. Consequently, it generates numerous terms, but in each term, on the $k$-th qubit, all possible situations that eventually arise are $\{X, Z; Z, X; Y, Z; Z, Y; X, I; I, X; Y, I; I, Y\}$. This implies that in $\vec{\sigma}_{i,0}^{k'_{2L}}, \vec{\sigma}_{j,0}^{k'_{2L}}$, there is at least one term with $X$ or $Y$. Additionally, since $\langle 0|X|0 \rangle = \langle 0|Y|0 \rangle = \langle 1|X|1 \rangle = \langle 1|Y|1 \rangle = 0$, it follows that $\text{Tr}[\vec{\sigma}_{i,0}^{k_{2L}} \rho_0] \text{Tr}[\vec{\sigma}_{j,0}^{k'_{2L}} \rho_0] = 0$. Therefore, we conclude that when $\sigma_{k,i,2L} = X$ and $\widetilde{\sigma}_{k,j,2L} = Z$, Eq. (96) equals 0.

In an analogous manner, when the initial Pauli matrix of the $k$-th qubit is $\{X, Y; Y, X; Y, Z; Z, X; Z, Y\}$, we can still obtain $\text{Tr}[\vec{\sigma}_{i,0}^{k_{2L}} \rho_0] \text{Tr}[\vec{\sigma}_{j,0}^{k'_{2L}} \rho_0] = 0$. Only when the initial state is one of $\{X, X; Y, Y; Z, Z; Z, I; I, Z; I, I\}$, $\text{Tr}[\vec{\sigma}_{i,0}^{k_{2L}} \rho_0] \text{Tr}[\vec{\sigma}_{j,0}^{k'_{2L}} \rho_0] \neq 0$. In light of the fact that both $O_i$ and $O_j$ are global observables, and $O_i \neq O_j$, it follows that there exists at least one position, such that the Pauli matrices at the $k$-th position of $O_i$ and $O_j$ belong to the set $\{X, Y; Y, X; Y, Z; Z, X; Z, Y\}$. Thus, for global observable operators $O_i$ and $O_j$, $\mathbb{E}_{\boldsymbol{\theta}} \left( \frac{\partial f_i(\boldsymbol{\theta})}{\partial \theta_{q,n}} \frac{\partial f_j(\boldsymbol{\theta})}{\partial \theta_{q,n}} \right) = 0$.

Following a similar analysis, we obtain $\mathbb{E}_{\boldsymbol{\theta}} \left( \frac{\partial f_{i_1}(\boldsymbol{\theta})}{\partial \theta_{q,n}} \frac{\partial f_{i_2}(\boldsymbol{\theta})}{\partial \theta_{q,n}} \right) = \mathbb{E}_{\boldsymbol{\theta}} \left( \frac{\partial f_{j_1}(\boldsymbol{\theta})}{\partial \theta_{q,n}} \frac{\partial f_{j_2}(\boldsymbol{\theta})}{\partial \theta_{q,n}} \right) = 0$. Thus, Eq. (89) can be simplified to:

$$\mathbb{E}_{\boldsymbol{\theta}} \|\nabla_{\boldsymbol{\theta}} f(\boldsymbol{\theta})\|^2 = \sum_{q,n,i} \mathbb{E}_{\boldsymbol{\theta}} \left( \frac{\partial f_i(\boldsymbol{\theta})}{\partial \theta_{q,n}} \right)^2 + \sum_{q,n,j} \mathbb{E}_{\boldsymbol{\theta}} \left( \frac{\partial f_j(\boldsymbol{\theta})}{\partial \theta_{q,n}} \right)^2 \tag{97}$$

$$\geq \sum_{q,n} \mathbb{E}_{\boldsymbol{\theta}} \left( \frac{\partial f_1(\boldsymbol{\theta})}{\partial \theta_{q,n}} \right)^2 \tag{98}$$

$$\geq \frac{1}{4} - \frac{1}{8L} \tag{99}$$

Thus, we have completed the proof of the lemma.

Next, let's proceed with the proof of Theorem 2. Without loss of generality, we select $O_1$ and initialize the parameters of the quantum circuit according to it. Next, we will expand $f(\theta)$ to obtain its expression:

$$\mathbb{E}_{\boldsymbol{\theta}} ||\nabla_{\boldsymbol{\theta}} f(\boldsymbol{\theta})||^2 = \sum_{q,n} \mathbb{E}_{\boldsymbol{\theta}} \left( \frac{\partial f(\boldsymbol{\theta})}{\partial \theta_{q,n}} \right)^2 \tag{100}$$

$$= \sum_{q,n} \mathbb{E}_{\boldsymbol{\theta}} \left( \sum_{i_1} \frac{\partial f'_{i_1}(\boldsymbol{\theta})}{\partial \theta_{q,n}} - \sum_{j_1} \frac{\partial f'_{j_1}(\boldsymbol{\theta})}{\partial \theta_{q,n}} + \sum_{i_2} \frac{\partial f''_{i_2}(\boldsymbol{\theta})}{\partial \theta_{q,n}} - \sum_{j_2} \frac{\partial f''_{j_2}(\boldsymbol{\theta})}{\partial \theta_{q,n}} \right)^2 \tag{101}$$

$$= \sum_{q,n} \mathbb{E}_{\boldsymbol{\theta}} \left( \sum_{i_1} \frac{\partial f'_{i_1}(\boldsymbol{\theta})}{\partial \theta_{q,n}} - \sum_{j_1} \frac{\partial f'_{j_1}(\boldsymbol{\theta})}{\partial \theta_{q,n}} \right)^2$$

$$+ 2 \sum_{q,n} \mathbb{E}_{\boldsymbol{\theta}} \left( \sum_{i_1} \frac{\partial f'_{i_1}(\boldsymbol{\theta})}{\partial \theta_{q,n}} - \sum_{j_1} \frac{\partial f'_{j_1}(\boldsymbol{\theta})}{\partial \theta_{q,n}} \right) \left( \sum_{i_2} \frac{\partial f''_{i_2}(\boldsymbol{\theta})}{\partial \theta_{q,n}} - \sum_{j_2} \frac{\partial f''_{j_2}(\boldsymbol{\theta})}{\partial \theta_{q,n}} \right)$$

$$+ \sum_{q,n} \mathbb{E}_{\boldsymbol{\theta}} \left( \sum_{i_2} \frac{\partial f''_{i_2}(\boldsymbol{\theta})}{\partial \theta_{q,n}} - \sum_{j_2} \frac{\partial f''_{j_2}(\boldsymbol{\theta})}{\partial \theta_{q,n}} \right)^2 \tag{102}$$

where $f'_{i_1}(\theta) = \mathrm{Tr}[\boldsymbol{O}'_{i_1} U(\theta)\rho_0 U(\theta)^\dagger]$, $f'_{j_1}(\theta) = \mathrm{Tr}[\boldsymbol{O}'_{j_1} U(\theta)\rho_0 U(\theta)^\dagger]$, $f''_{i_2}(\theta) = \mathrm{Tr}[\boldsymbol{O}''_{i_2} U(\theta)\rho_0 U(\theta)^\dagger]$, $f''_{j_2}(\theta) = \mathrm{Tr}[\boldsymbol{O}''_{j_2} U(\theta)\rho_0 U(\theta)^\dagger]$. The notations $\boldsymbol{O}'_{i_1}$ and $\boldsymbol{O}'_{j_1}$ suggest that, in comparison to $\boldsymbol{O}_1$, they simply involve replacing some Pauli matrices Z with I or vice versa. For instance, consider $X \otimes Y \otimes Z \otimes I$ and $X \otimes Y \otimes I \otimes Z$. On the other hand, $\boldsymbol{O}''_{i_2}, \boldsymbol{O}''_{j_2}$ represent other observables.

Following similar analyses from Lemma 6, we determine that the second term in Eq. 102 is equal to 0. Now, let's expand the remaining terms. Therefore:

$$\mathbb{E}_{\boldsymbol{\theta}} ||\nabla_{\boldsymbol{\theta}} f(\boldsymbol{\theta})||^2 = \sum_{q,n} \mathbb{E}_{\boldsymbol{\theta}} \left( (\sum_{i_1} \frac{\partial f'_{i_1}(\boldsymbol{\theta})}{\partial \theta_{q,n}} - \sum_{j_1} \frac{\partial f'_{j_1}(\boldsymbol{\theta})}{\partial \theta_{q,n}})^2 + (\sum_{i_2} \frac{\partial f''_{i_2}(\boldsymbol{\theta})}{\partial \theta_{q,n}} - \sum_{j_2} \frac{\partial f''_{j_2}(\boldsymbol{\theta})}{\partial \theta_{q,n}})^2 \right) \tag{103}$$

$$\geq \sum_{q,n} \mathbb{E}_{\boldsymbol{\theta}} \left( \sum_{i_1} \frac{\partial f'_{i_1}(\boldsymbol{\theta})}{\partial \theta_{q,n}} - \sum_{j_1} \frac{\partial f'_{j_1}(\boldsymbol{\theta})}{\partial \theta_{q,n}} \right)^2 \tag{104}$$

$$= \sum_{q,n,i_1} \mathbb{E}_{\boldsymbol{\theta}} \left( \frac{\partial f'_{i_1}(\boldsymbol{\theta})}{\partial \theta_{q,n}} \right)^2 + \sum_{q,n,i'_1 \neq i''_1} \mathbb{E}_{\boldsymbol{\theta}} \left( \frac{\partial f'_{i'_1}(\boldsymbol{\theta})}{\partial \theta_{q,n}} \cdot \frac{\partial f'_{i''_1}(\boldsymbol{\theta})}{\partial \theta_{q,n}} \right)$$

$$- 2 \sum_{q,n,i_1,j_1} \mathbb{E}_{\boldsymbol{\theta}} \left( \frac{\partial f'_{i_1}(\boldsymbol{\theta})}{\partial \theta_{q,n}} \cdot \frac{\partial f'_{j_1}(\boldsymbol{\theta})}{\partial \theta_{q,n}} \right)$$

$$+ \sum_{q,n,j_1} \mathbb{E}_{\boldsymbol{\theta}} \left( \frac{\partial f'_{j_1}(\boldsymbol{\theta})}{\partial \theta_{q,n}} \right)^2 + \sum_{q,n,j'_1 \neq j''_1} \mathbb{E}_{\boldsymbol{\theta}} \left( \frac{\partial f'_{j'_1}(\boldsymbol{\theta})}{\partial \theta_{q,n}} \cdot \frac{\partial f'_{j''_1}(\boldsymbol{\theta})}{\partial \theta_{q,n}} \right) \tag{105}$$

It is easy to see that all the cross terms in this expression differ in the positions where $I$ and $Z$ occur. Therefore, there exists a $k$ such that the $k$-th position in $f'_{i_1}(\boldsymbol{\theta})$ and $f'_{j_1}(\boldsymbol{\theta})$ is either $I, Z$ or $Z, I$. According to Eq. (42) and Eq. (44), we know that the third term in Eq. (105) is equal to 0. Similarly, we can analyze the other cross terms in Eq. (105) and conclude that they are all equal to 0. Therefore, we have:

$$\mathbb{E}_{\boldsymbol{\theta}} ||\nabla_{\boldsymbol{\theta}} f(\boldsymbol{\theta})||^2 \geq \sum_{q,n,i_1} \mathbb{E}_{\boldsymbol{\theta}} \left( \frac{\partial f'_{i_1}(\boldsymbol{\theta})}{\partial \theta_{q,n}} \right)^2 + \sum_{q,n,j_1} \mathbb{E}_{\boldsymbol{\theta}} \left( \frac{\partial f'_{j_1}(\boldsymbol{\theta})}{\partial \theta_{q,n}} \right)^2 \tag{106}$$

Given that $\boldsymbol{O}'_{i_1}$ and $\boldsymbol{O}_1$ differ only in certain terms that flip $I$ to $Z$ or $Z$ to $I$, and during the initialization of quantum circuit parameters, the $k$-th position in $\boldsymbol{O_1}$ follows $\mathcal{G}_3(\sigma^2)$ if it is $I$ or $Z$. Therefore, for all $i_1$, $\sum_{q,n} \mathbb{E}_{\boldsymbol{\theta}} \left( \frac{\partial f'_{i_1}(\boldsymbol{\theta})}{\partial \theta_{q,n}} \right)^2$ are all equal. According to Eq. (50) and Eq. (51), and employing a similar analysis to Theorem 1, we obtain:

$$\sum_{q,n} \mathbb{E}_{\boldsymbol{\theta}} \left( \frac{\partial f'_{i_1}(\boldsymbol{\theta})}{\partial \theta_{q,n}} \right)^2 \geq \frac{1}{4} - \frac{1}{8L} \tag{107}$$

Thus, we have:

$$\mathbb{E}_{\boldsymbol{\theta}} \|\nabla_{\boldsymbol{\theta}} f(\boldsymbol{\theta})\|^2 \geq M \left( \frac{1}{4} - \frac{1}{8L} \right) \tag{108}$$

### A.4 PROOF OF THEOREM 3

Without loss of generality, we select $\boldsymbol{O}_1$ and initialize according to $\boldsymbol{O}_1$. Let $\boldsymbol{O}_1 = o_1^1 \otimes o_2^1 \otimes ... \otimes o_N^1$. We expand $f(\boldsymbol{\theta})$ to obtain:

$$\mathbb{E}_{\boldsymbol{\theta}} \|\nabla_{\boldsymbol{\theta}} f(\boldsymbol{\theta})\|^2 = \sum_{q,n} \mathbb{E}_{\boldsymbol{\theta}} \left( \frac{\partial f(\boldsymbol{\theta})}{\partial \theta_{q,n}} \right)^2 \tag{109}$$

$$= \sum_{q,n} \mathbb{E}_{\boldsymbol{\theta}} \left( \sum_i \frac{\partial f'_i(\boldsymbol{\theta})}{\partial \theta_{q,n}} + \sum_j \frac{\partial f''_j(\boldsymbol{\theta})}{\partial \theta_{q,n}} \right)^2 \tag{110}$$

$$= \sum_{q,n} \mathbb{E}_{\boldsymbol{\theta}} \left( \sum_i \frac{\partial f'_i(\boldsymbol{\theta})}{\partial \theta_{q,n}} \right)^2 + 2 \sum_{q,n} \mathbb{E}_{\boldsymbol{\theta}} \left( \sum_{i,j} \frac{\partial f'_i(\boldsymbol{\theta})}{\partial \theta_{q,n}} \cdot \frac{\partial f''_j(\boldsymbol{\theta})}{\partial \theta_{q,n}} \right) + \sum_{q,n} \mathbb{E}_{\boldsymbol{\theta}} \left( \sum_j \frac{\partial f''_j(\boldsymbol{\theta})}{\partial \theta_{q,n}} \right)^2 \tag{111}$$

$$\geq \sum_{q,n,i} \mathbb{E}_{\boldsymbol{\theta}} \left( \frac{\partial f'_i(\boldsymbol{\theta})}{\partial \theta_{q,n}} \right)^2 + \sum_{q,n,i_1 \neq i_2} \mathbb{E}_{\boldsymbol{\theta}} \left( \frac{\partial f'_{i_1}(\boldsymbol{\theta})}{\partial \theta_{q,n}} \cdot \frac{\partial f'_{i_2}(\boldsymbol{\theta})}{\partial \theta_{q,n}} \right) + 2 \sum_{q,n,i,j} \mathbb{E}_{\boldsymbol{\theta}} \left( \frac{\partial f'_i(\boldsymbol{\theta})}{\partial \theta_{q,n}} \cdot \frac{\partial f''_j(\boldsymbol{\theta})}{\partial \theta_{q,n}} \right) \tag{112}$$

where $f'_i(\boldsymbol{\theta}) = \mathrm{Tr}[\boldsymbol{O}'_i U(\boldsymbol{\theta}) \rho_0 U(\boldsymbol{\theta})^\dagger]$ and $f''_j(\boldsymbol{\theta}) = \mathrm{Tr}[\boldsymbol{O}'_j U(\boldsymbol{\theta}) \rho_0 U(\boldsymbol{\theta})^\dagger]$. $\boldsymbol{O}'_i$ implies that, compared to $\boldsymbol{O}_1$, they might have operations that flip some $I$ to $Z$ or $Z$ to $I$, while the rest of the Pauli matrices are the same. $\boldsymbol{O}'_j$ represents observables that do not satisfy these conditions.

According to a similar analysis as in Lemma 6, we can see that the third term in Eq. (112) is equal to 0. In the context of the final block, where the positions of $I$ and $Z$ in $\boldsymbol{O}_1$ follow Gaussian distributions $\mathcal{N}(0, \sigma^2)$, and considering that $\boldsymbol{O}'_i$, compared to $\boldsymbol{O}_1$, only involves flipping Pauli I to Pauli Z or Pauli Z to Pauli I, we can apply a similar analysis as in Theorem 1. As a result, in the first term of Eq. (112), for each $\boldsymbol{O}'_i$, we find that $\sum_{q,n} \mathbb{E}_{\boldsymbol{\theta}} \left( \frac{\partial f'_i(\boldsymbol{\theta})}{\partial \theta_{q,n}} \right)^2 \geq \frac{1}{4} - \frac{1}{8L}$. For the second term in Eq. (112), when $n \in P_{1:3}^{ij}$ and $q \in [2L-2]$, note that:

$$\mathbb{E}_{\boldsymbol{\theta}}\left(\frac{\partial f'_{i_1}(\boldsymbol{\theta})}{\partial \theta_{q,n}}\frac{\partial f'_{i_2}(\boldsymbol{\theta})}{\partial \theta_{q,n}}\right)$$

$$= \mathbb{E}_{\boldsymbol{\theta}}\left(\frac{\partial}{\partial \theta_{q,n}}\text{Tr}[\boldsymbol{O}'_{i_1}\rho_{2L}]\frac{\partial}{\partial \theta_{q,n}}\text{Tr}[\boldsymbol{O}'_{i_2}\rho_{2L}]\right) \tag{113}$$

$$= \mathbb{E}_{\boldsymbol{\theta}_1}\cdots\mathbb{E}_{\boldsymbol{\theta}_{2L}}\left(\frac{\partial}{\partial \theta_{q,n}}\text{Tr}[\boldsymbol{O}'_{i_1}R_{2L}(\boldsymbol{\theta}_{2L})\rho_{2L-1}R^\dagger_{2L}(\boldsymbol{\theta}_{2L})]\frac{\partial}{\partial \theta_{q,n}}\text{Tr}[\boldsymbol{O}'_{i_2}R_{2L}(\boldsymbol{\theta}_{2L})\rho_{2L-1}R^\dagger_{2L}(\boldsymbol{\theta}_{2L})]\right) \tag{114}$$

$$\geq \alpha^{S_1^{i_1 i_2}+S_3^{i_1 i_2}}\gamma^{S_{0,3}^{i_1 i_2}}\mathbb{E}_{\boldsymbol{\theta}_1}\cdots\mathbb{E}_{\boldsymbol{\theta}_{2L-1}}\left(\frac{\partial}{\partial \theta_{q,n}}\text{Tr}[\boldsymbol{O}'_{3:i_1;1}\rho_{2L-1}]\frac{\partial}{\partial \theta_{q,n}}\text{Tr}[\boldsymbol{O}'_{3:i_2;1}\rho_{2L-1}]\right) \tag{115}$$

$$\geq \alpha^{S_1^{i_1 i_2}+S_3^{i_1 i_2}}\gamma^{S_{0,3}^{i_1 i_2}}\mathbb{E}_{\boldsymbol{\theta}_1}\cdots\mathbb{E}_{\boldsymbol{\theta}_{2L-1}}\left(\frac{\partial}{\partial \theta_{q,n}}\text{Tr}[\boldsymbol{O}'_{3:i_1;1}R_{2L-1}(\boldsymbol{\theta}_{2L-1})CZ_L\rho_{2L-2}CZ_L^\dagger R^\dagger_{2L-1}(\boldsymbol{\theta}_{2L-1})]\right.$$

$$\left.\frac{\partial}{\partial \theta_{q,n}}\text{Tr}[\boldsymbol{O}'_{3:i_2;1}R_{2L-1}(\boldsymbol{\theta}_{2L-1})CZ_L\rho_{2L-2}CZ_L^\dagger R^\dagger_{2L-1}(\boldsymbol{\theta}_{2L-1})]\right) \tag{116}$$

$$\geq \alpha^{S_1^{i_1 i_2}+S_3^{i_1 i_2}}\alpha^{S_{1:3}^{i_1 i_2}}\gamma^{2S_{0,3}^{i_1 i_2}}\mathbb{E}_{\boldsymbol{\theta}_1}\cdots\mathbb{E}_{\boldsymbol{\theta}_{2L-2}}\left(\frac{\partial}{\partial \theta_{q,n}}\text{Tr}[\boldsymbol{O}'_{3:i_1}CZ_L\rho_{2L-2}CZ_L^\dagger]\frac{\partial}{\partial \theta_{q,n}}\text{Tr}[\boldsymbol{O}'_{3:i_2}CZ_L\rho_{2L-2}CZ_L^\dagger]\right) \tag{117}$$

$$= \alpha^{S_1^{i_1 i_2}+S_3^{i_1 i_2}}\alpha^{S_{1:3}^{i_1 i_2}}\gamma^{2S_{0,3}^{i_1 i_2}}\mathbb{E}_{\boldsymbol{\theta}_1}\cdots\mathbb{E}_{\boldsymbol{\theta}_{2L-2}}\left(\frac{\partial}{\partial \theta_{q,n}}\text{Tr}[\boldsymbol{O}'_{3:i_1}\rho_{2L-2}]\frac{\partial}{\partial \theta_{q,n}}\text{Tr}[\boldsymbol{O}'_{3:i_2}\rho_{2L-2}]\right) \tag{118}$$

$$\geq \alpha^{S_1^{i_1 i_2}+S_3^{i_1 i_2}}\alpha^{(2L-q-1)S_{1:3}^{i_1 i_2}}\gamma^{(2L-q)S_{0,3}^{i_1 i_2}}\mathbb{E}_{\boldsymbol{\theta}_1}\cdots\mathbb{E}_{\boldsymbol{\theta}_q}\left(\frac{\partial}{\partial \theta_{q,n}}\text{Tr}[\boldsymbol{O}'_{3:i_1}\rho_q]\frac{\partial}{\partial \theta_{q,n}}\text{Tr}[\boldsymbol{O}'_{3:i_2}\rho_q]\right) \tag{119}$$

Similar to Eq. (73), Eq. (115) is derived from Eq. (17), (23), (50) and (52). Similarly, we obtain Eq. (117). Eq. (118) is simplified through Lemma 5. Continuing this analysis up to layer $q$, we arrive at Eq. (119).

$$\mathbb{E}_{\boldsymbol{\theta}}\left(\frac{\partial f'_{i_1}(\boldsymbol{\theta})}{\partial \theta_{q,n}}\frac{\partial f'_{i_2}(\boldsymbol{\theta})}{\partial \theta_{q,n}}\right)$$

$$= \mathbb{E}_{\boldsymbol{\theta}}\left(\frac{\partial}{\partial \theta_{q,n}}\text{Tr}[\boldsymbol{O}'_{i_1}\rho_{2L}]\frac{\partial}{\partial \theta_{q,n}}\text{Tr}[\boldsymbol{O}'_{i_2}\rho_{2L}]\right) \tag{120}$$

$$\geq \alpha^{S_1^{i_1 i_2}+S_3^{i_1 i_2}}\alpha^{(2L-q-1)S_{1:3}^{i_1 i_2}}\gamma^{(2L-q+1)S_{0,3}^{i_1 i_2}}\alpha^{S_{1:3}^{i_1 i_2}-1}\beta\mathbb{E}_{\boldsymbol{\theta}_1}\cdots\mathbb{E}_{\boldsymbol{\theta}_{q-1}}\left(\text{Tr}[\boldsymbol{O}'_{3:i_1}\rho_{q-1}]\text{Tr}[\boldsymbol{O}'_{3:i_2}\rho_{q-1}]\right) \tag{121}$$

$$\geq \alpha^{S_1^{i_1 i_2}+S_3^{i_1 i_2}}\alpha^{(2L-1)S_{1:3}^{i_1 i_2}}\gamma^{2LS_{0,3}^{i_1 i_2}}\beta\text{Tr}[\boldsymbol{O}'_{3:i_1}\rho_0]\text{Tr}[\boldsymbol{O}'_{3:i_2}\rho_0] \tag{122}$$

$$\geq \alpha^{2LS_{1:3}^{i_1 i_2}-1}\gamma^{2LS_{0,3}^{i_1 i_2}}\beta \tag{123}$$

$$\geq (1-\sigma^2)^{2LS_{1:3}^{i_1 i_2}-1}e^{-L\sigma^2 S_{0,3}^{i_1 i_2}}\sigma^2(1-\sigma^2) \tag{124}$$

$$= \frac{1}{2LS}\left(1-\frac{1}{2LS}\right)^{2LS_{1:3}^{i_1 i_2}}e^{-\frac{S_{0,3}^{i_1 i_2}}{2S}}, \tag{125}$$

Eq. (120) to Eq. (125) follow a similar analysis to Eq. (78) and Eq. (82). When $n \in P_{1:3}^{ij}$, a similar analysis reveals that when $q = 2L - 1$,

$$\mathbb{E}_{\boldsymbol{\theta}}\left(\frac{\partial f(\boldsymbol{\theta})}{\partial \theta_{q,n}}\right)^2 \geq \frac{1}{2LS}(1-\frac{1}{2LS})^{2LS_{1:3}^{i_1 i_2}}e^{-\frac{S_{0,3}^{i_1 i_2}}{2S}}, \tag{126}$$

and when $q = 2L$, $\mathbb{E}_{\boldsymbol{\theta}}(\frac{\partial f(\boldsymbol{\theta})}{\partial \theta_{q,n}})^2 \geq 0$. Fig. 6 and 7 illustrate the evolution of the first cross-terms in Eq. 112 for different configurations of Pauli matrices at each position. According to Lemma 5,

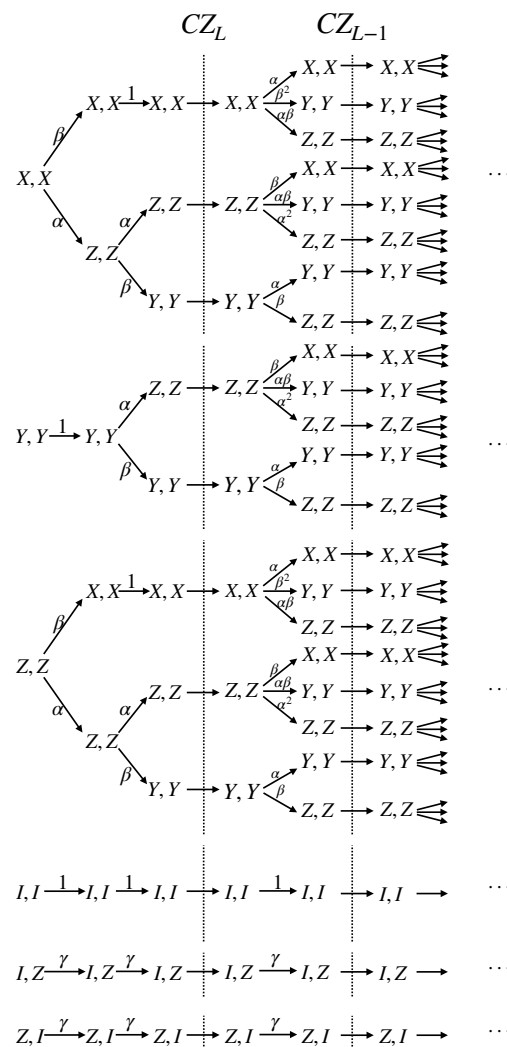

Figure 6: At each position, depending on the different initial Pauli matrices, various terms are generated. This indicates that when the initial Pauli matrix at any position belongs to $\{XX, YY, ZZ, II, IZ, ZI\}$, it shows the transformation of the Pauli matrix and the corresponding coefficients. When the Pauli matrix undergoes a CZ gate, according to Lemma 5, it may involve flipping operations. Here, it illustrates the scenario when no flips exist, showcasing the changes in the Pauli matrix. Here, the dashed line to the left of $CZ_L$ represents the change in different Pauli matrices as they pass through the two rotation gates in the $L$-th block. The transition from $CZ_L$ to $CZ_{L-1}$ indicates the transformation of the Pauli matrices as they pass through the $(L-1)$-th block.

$CZ$ may execute a flip operation. Therefore, we discuss two scenarios: one where no flip occurs, as shown in Fig. 6, and another where $CZ$ causes a flip of Pauli matrices, as depicted in Fig 7. As mentioned earlier, we find that if the k-th Pauli matrix is to undergo a flip operation, we require the (k-1)-th position to have a Pauli matrix of $X$ or $Y$, and the (k+1)-th position to have a Pauli matrix of $Z$ or $I$, or vice versa. Taking into account that some terms in the evolution of $iGO\rho$ may yield coefficients with negative signs, our specific setup ensures that when the coefficient for the preceding Pauli matrix becomes negative, the succeeding Pauli matrix will also inevitably have a negative coefficient. Consequently, the final coefficients are positive. When $n \notin P_{1:3}^{ij}$, i.e., $n \in P_0^{ij}$, we can easily deduce that $\mathbb{E}_{\boldsymbol{\theta}}(\frac{\partial f(\theta)}{\partial \theta_{q,n}})^2 \geq 0$. In conclusion, we can draw the following conclusions:

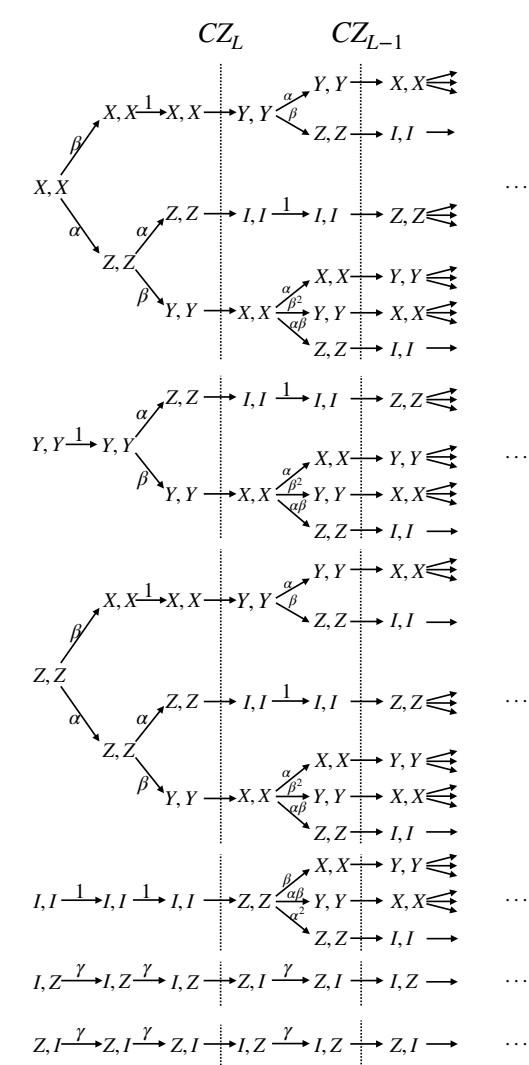

Figure 7: As before, it illustrates changes in the Pauli matrix. However, in this case, we assume that the CZ gate introduces flip operations.

$$\underset{\boldsymbol{\theta}}{\mathbb{E}}\,||\nabla_{\boldsymbol{\theta}}f(\boldsymbol{\theta})||_2^2 \geq M(\frac{1}{4}-\frac{1}{8L}) + \sum_{i\neq j=1}^{M}\frac{(2L-1)S_3^{ij}}{2LS}(1-\frac{1}{2LS})^{2LS_{1:3}^{ij}}e^{-\frac{S_{0,3}^{ij}}{2S}} + \sum_{q,n,j}E\left(\frac{\partial f_j''(\boldsymbol{\theta})}{\partial\theta_{q,n}}\right)^2 \tag{127}$$

$$\geq M(\frac{1}{4}-\frac{1}{8L}) + \sum_{i\neq j=1}^{M}\frac{(2L-1)S_3^{ij}}{2LS}(1-\frac{1}{2LS})^{2LS_{1:3}^{ij}}e^{-\frac{S_{0,3}^{ij}}{2S}} \tag{128}$$

## B  SIMULATED EXPERIMENTS IN QUANTUM CHEMISTRY

In the following, we explore the application of our initialization method to compute the ground-state energy of the LiH molecule, a benchmark in quantum chemistry. Its loss function is global. For an electronic system with $N$ electrons distributed over $M$ spin molecular orbitals, the initial state is the Hartree-Fock (HF) state:

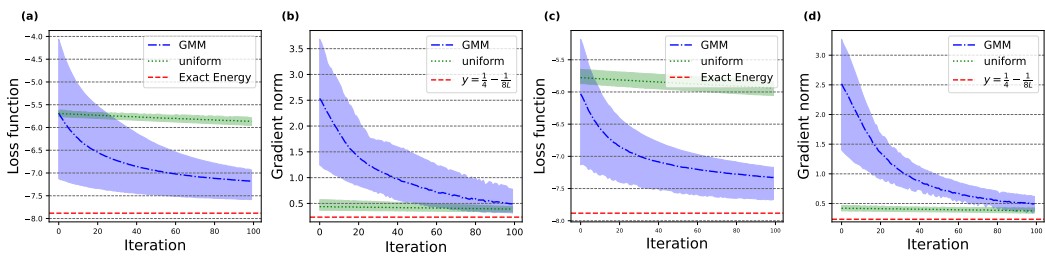

Figure 8: When $L = 10$, we examine the variation of the cost function and $\underset{\boldsymbol{\theta}}{\mathbb{E}} ||\nabla_{\boldsymbol{\theta}} f(\boldsymbol{\theta})||^2$ under noisy and noise-free conditions, using both uniform distribution $(\mathcal{U}[-\pi, \pi])$ and GMM-initialized parameters. Where (a) and (b) represent the noise-free scenario, while (c) and (d) represent the case with noise.

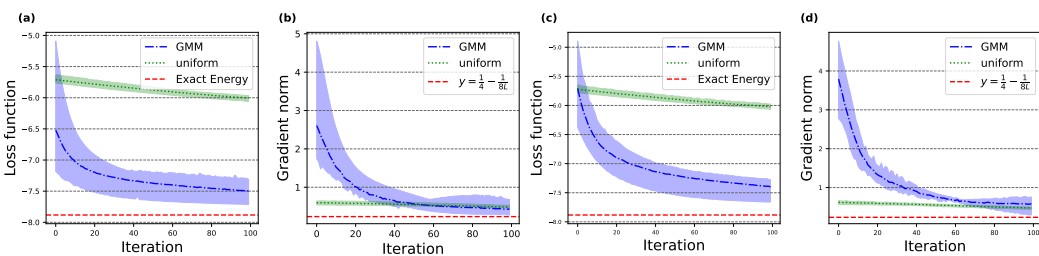

Figure 9: When $L = 20$, (a) and (c) depict the loss function under noise-free and noisy conditions, respectively, with a uniform distribution $(\mathcal{U}[-\pi, \pi])$ and GMM-initialized parameters. On the other hand, (b) and (d) illustrate the changes in $\underset{\boldsymbol{\theta}}{\mathbb{E}} ||\nabla_{\boldsymbol{\theta}} f(\boldsymbol{\theta})||^2$ under noise-free and noisy conditions, respectively.

$$|\Phi\rangle_{HF} = |\overbrace{11...11}^{N}\underbrace{00...00}_{M}\rangle.$$

In the LiH molecule, with an electron count of $N = 2$ and $M = 10$ free spin orbitals, simulating electronic structure problems on a quantum computer requires establishing a mapping that transforms fermionic operators of electrons into Pauli operators. Common mappings include the Jordan-Wigner (JW) transformation, Bravyi-Kitaev (BK) transformation, and Parity transformation. Here, we adopt the JW mapping to compute its ground-state energy.

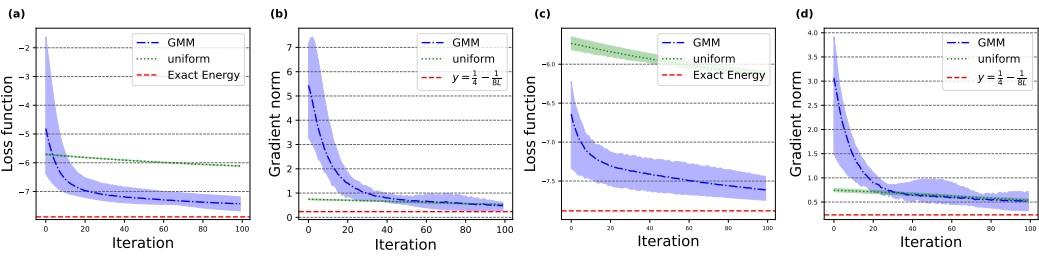

Figure 10: When $L = 30$, (a) illustrates the variation of the loss under noise-free conditions; (b) depicts $\underset{\boldsymbol{\theta}}{\mathbb{E}} ||\nabla_{\boldsymbol{\theta}} f(\boldsymbol{\theta})||^2$ under noise-free conditions; (c) shows the change in loss under noisy conditions; and (d) displays $\underset{\boldsymbol{\theta}}{\mathbb{E}} ||\nabla_{\boldsymbol{\theta}} f(\boldsymbol{\theta})||^2$ under noisy conditions.

We set the number of layers ($L$) to 10, 20, and 30, using a gradient descent optimizer with a learning rate of 0.01. Additionally, we consider the impact of the noise on the barren plateau problem by introducing a moderate amount of noise during training to simulate real-world quantum computer operation. We compare the evolution of the cost function and $\mathbb{E}_{\boldsymbol{\theta}} ||\nabla_{\boldsymbol{\theta}} f(\boldsymbol{\theta})||^2$ during training when initializing parameters using GMM and uniform distribution $\mathcal{U}[-\pi, \pi]$. The results are shown in Fig. 8, 9, and 10. In each figure, (a) and (b) represent the condition without noise, while (c) and (d) represent the noisy condition. From the results, we observe that regardless of the value of $L$ or the presence of noise, initializing parameters using the GMM method consistently provides a larger $\mathbb{E}_{\boldsymbol{\theta}} ||\nabla_{\boldsymbol{\theta}} f(\boldsymbol{\theta})||^2$ at the beginning of training and it consistently stays much higher than the lower bound we have provided. This value remains relatively high before the convergence of the cost function, therefore, the GMM initialization ensures a rapid convergence. On the other hand, the uniform distribution $\mathcal{U}[-\pi, \pi]$ maintains a consistently lower level of gradient norm, resulting in a significantly slower convergence process.

Next, let's consider the impact of the parameter $\sigma^2$ in the GMM. In the main text, we set $\sigma^2$ to be $\frac{1}{2LS}$. We compare the training scenarios with different $\sigma^2$ values under noisy and noise-free conditions when $L = 10, 20, 30$. Here, $\sigma^2$ is chosen as $0.1 \times \frac{1}{2LS}$, $\frac{1}{2LS}$, and $10 \times \frac{1}{2LS}$. The results are shown in Fig. 11, 12, and 13.

As before, (a) and (b) represent noise-free conditions, while (c) and (d) represent scenarios with noise. The results in the figures indicate that when $\sigma^2 = 10 \times \frac{1}{2LS}$, the convergence of the cost function is significantly slower. On the other hand, when $\sigma^2 = 0.1 \times \frac{1}{2LS}$, although the cost function converges, its results are often inferior to the original case, especially in the presence of noise. We believe that as $\sigma^2$ increases, the peaks of the probability density function in the GMM become lower, and its distribution becomes closer to the uniform distribution, leading to a smaller KL divergence between them. Conversely, when $\sigma^2$ decreases, the peaks of the GMM's probability density function become higher. Therefore, the data becomes more concentrated around the peaks, making it less dispersed. This may be the reason why the convergence results are not as good as when $\sigma^2 = \frac{1}{2LS}$.

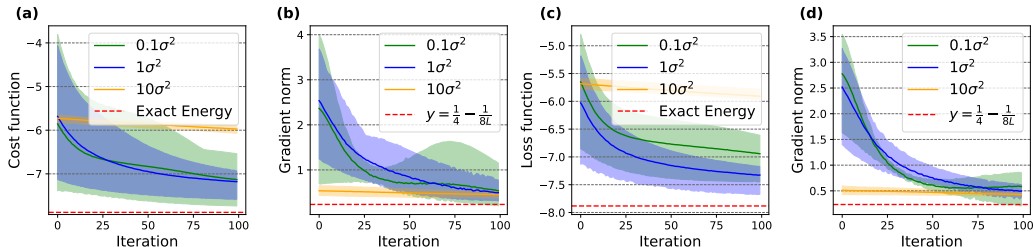

Figure 11: In the configuration with $L = 10$, the impact of different $\sigma^2$ on training under noisy and noise-free conditions is depicted. Here, (a) and (b) represent the noise-free scenario, while (c) and (d) represent the noisy scenario.

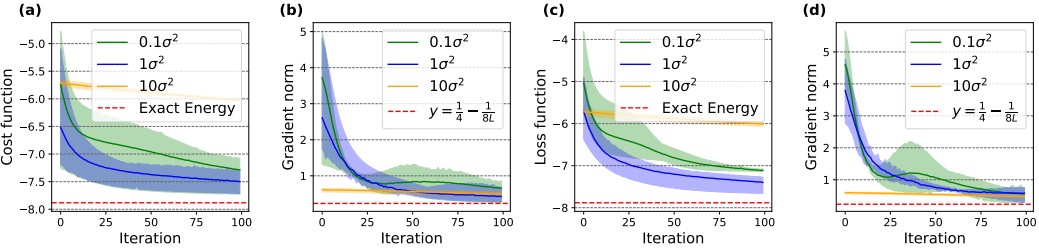

Figure 12: For a 20-layer configuration, the impact of different $\sigma^2$ on training under noisy and noise-free conditions is depicted. Here, (a) and (b) represent the noise-free scenario, while (c) and (d) represent the noisy situation.

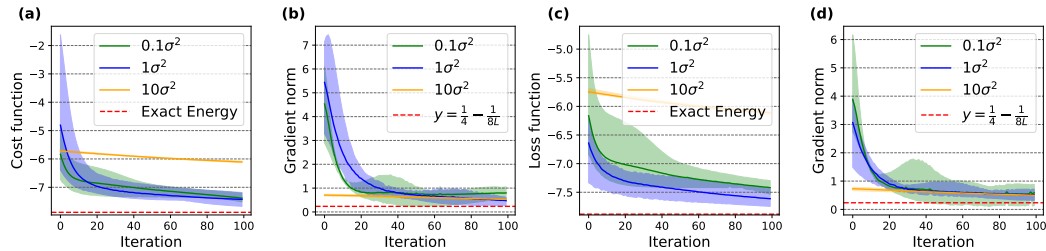

Figure 13: In the $L = 30$ configuration, (a) and (b) illustrate the impact of different $\sigma^2$ on training under noise-free conditions, while (c) and (d) depict the influence of various $\sigma^2$ under noisy conditions.

