# OpenReview forum: "AVOIDING BARREN PLATEAUS VIA GAUSSIAN MIXTURE MODEL"
_ICLR.cc/2025/Conference — ICLR 2025 Conference Withdrawn Submission_

### Official Review · Reviewer_C1Ja · 2024-10-25

**Soundness:** 1
**Presentation:** 1
**Contribution:** 1
**Rating:** 3
**Confidence:** 3

**Summary:**

Variational Quantum Algorithms (VQAs) are important tools to exploit the capabilities of Noisy Intermediate Scale Quantum devices. The basic idea is to leverage parametrized quantum circuits (PQCs), whose parameters are often angles of rotational gates, to find the ground state of a given Hamiltonian, which plays the role of the cost function.

VQAs, however, suffer from several shortcomings. One of the most sever one is known as Barren Plateaus (BPs). When dealing with many qubits and layers in the quantum circuit, the number of variational parameters rapidly increases thus making the optmization problem more challenging. Furthermore, the optimization landscape becomes extrmely difficult to navigate and this results in very small gradient signals (during the optimization of the parametrized quantum circuits) which prevents to converge to global minima and, therefore, to the desired ground state wave function.

In this paper the authors propose to use Gaussian Mixture Models (GMMs) for the initialization of the parameters in PQCs.  The main contribution of the paper claims to **rigorously** solve the problem of **barren plateaus** by initializing the parameters of PQCs which in turn leads to higher gradient signals throughout the optimization of the parameters.

**Strengths:**

- The paper adresses a fundamental problem in the context of variational quantum algoritms.

**Weaknesses:**

- The major weakness of this paper lies in its claim. In the abstract the authors claim that "*rigorously prove that, the proposed initialization method consistently avoids the barren plateaus problem for hardware-efficient ansatz*". While this may even be true in theory, I do find their numerical experiments and the PQCs considered in the paper not to be general such that this claim can hold with the current phrasing. This very **strong** claim is reiterated several times throughout the paper. I strongly advise to motigate such claim to something more aligned with the results of the manuscript.
- Another important weakness of the paper is that the code for reproducing the experiments is not provided thus preventing reproducibility of the experiments and further investigation of the implementation.
- Furthermore, I find the presentation of the paper hard to follow at times. The notation is often confusing cluttered. I believe that recalling what each variable refers to, use more display math (instead of inline equations) and provide some intuitive sketches may help.
- The ansatzes used in the paper are not general and do not expliclty account for entanglement. As it is known in the field of quantum computing, a set of universal gates, e.g., a collection of gates able to represent any possible unitary transformation, consist of the rotational gates and entangling gates such as CNOT. I believe this aspect substantially limits the generizability of the proposed claims and theorems. Does the set of gates considered in this work represents a set of universal gates? If yes that should be made clear.
- I find the numerical results shown in the main paper to be somewhat inconclusive. In particular, the authors omit many fundamental details such as the type of algorithm used to optimize the PQC, if it consists of global or local optimizations at each iteration, if measurement noise and/or hardware noise are taken into account for these experiments, whether the results change when with different algorithms etc. I believe that a thorough ablation study is necessary in order to support the strong claims of this work.
- I furthermore find puzzling that other results on quantum chemistry simulation are limited to the appendix as I believe those may arguably be even more relevant (or at least complementary) compared to the Ising model.
- About the layout of the paper, it looks to me as if Table 2 and Table 3, as well as Table 5 and Table 6 are duplicates of each other. Could it be, or am I missing something?
- Another concern about the layout of the paper is about the citation style. I strongly recommend to adopt fix the citation styles, e.g., substituting \cite with \citep where needed in order to wrap refs around parenthesis where suitable.
- The notion of "observable" may not be immediate to grasp for the general audience, I believe it would be useful to provide more concrete examples of real physical examples which could be mapped onto these generalized $\mathbf{\mathcal{O}}$ discussed in the paper.
- The writing and the clarity of the paper I think can in general be improved.
- Table 4: I think it would be good to mention that higher gradients at initialization is better. This may not be intuitive at first sight. Perhaps making the best results in bold without  editing the caption would already help.
- Line 32: I find the claim "*[...] VQAs provides a feasible approach to solving complex problems [...]*" to be far too general. I strongly encourage to be more specific about what VQA can be good at.
- Line 83: I recommend to change the wording *complex distribution* to complicated/non-trivial as the former may be confused to have different meaning, e.g., distribution of complex values.
- I storngly recommend to add a "Related work" section to give more structure to the paper and streamline the reading. Furthermore, I'd recommend a more thorough review citing also other work using ML methods to enhance optimization of PQCs such as Refs. [1-3] below. Similarly, I recommend to provide the list of contributions at the bottom of page 2 in bulletpoints so that they end up being more accessible and more evident to readers.
- For the people non familiar with optimization of PQCs, I'd briefly introduce the parameter shift rule and how to compute gradients on quantum computers at the begining of the last paragraph before section 2. I believe this would be useful to make the paper self-contained.
- Line 158: Something seems wrong with the parenthesis in the last equation of the sentence. As mentioned above, I strongly recommend to use more dispay math instead of inline equations which are often cluttered and hard to parse. If the authors need space I'd recommend removing one of the two figures Figure 1 and Figure 2 as I think the key messages therein can be merged into one figure.
- On the other hand I'd find beneficial to have some intuitive sketch about the main results/theorem the authors claim in the paper. That might make the paper more accessible also to a general audience (more from the ML community). At the moment the paper seems very much suited to an audience of physicists. For instance the authors never define what's a pure state. This cannot be assumed as common knowledge in the broader audience targeting this conference.
- To streamline the reading of the paper I'd find it useful to often recall what $q,n$ and $L,N,M$ are. That would help a lot to navigate both theorems and follow the sketch of proofs.
- Why does the gradient norm shown in figure 4 (right panel) has this double peak structure? Does this has any physical meaning? Is it intuitive why someone should expect such a steep increase in gradient norm during optimization? I believe this associates to the capability of the proposed algorithms to overcome barren plateaus but this is not discussed explicitly neither in the caption nor in the text. This might make it hard to the reader not familiar with the problem of Barren Plateaus to immediately grasp this.
- In line 470: "*We validate our algorithm for diverse problems, [...]*" I think this is not entirely correct. The paper only tackles (in the main part) the Transverse Field Ising Model with different setup. I think the authors should be clearer and more explicit here. This comment often applies to other parts in the paper where it would be useful to revisit the paper in ordare to ensure more precise claims.
- In line 483: what does the HEA acronym mean?

### References

- [1] [Tamiya, Shiro, and Hayata Yamasaki. "Stochastic gradient line Bayesian optimization for efficient noise-robust optimization of parameterized quantum circuits." npj Quantum Information 8.1 (2022): 90.](https://www.nature.com/articles/s41534-022-00592-6)
- [2] [Nicoli, Kim, et al. "Physics-informed bayesian optimization of variational quantum circuits." Advances in Neural Information Processing Systems 36 (2024).](https://proceedings.neurips.cc/paper_files/paper/2023/file/3adb85a348a18cdd74ce99fbbab20301-Paper-Conference.pdf)
- [3] [Anders, Christopher J., et al. "Adaptive Observation Cost Control for Variational Quantum Eigensolvers." Forty-first International Conference on Machine Learning.](https://openreview.net/pdf?id=dSrdnhLS2h)

**Questions:**

Please refer to the section above as often question associates to the weaknesses of the paper.

---

### Official Review · Reviewer_aycC · 2024-11-01

**Soundness:** 3
**Presentation:** 2
**Contribution:** 2
**Rating:** 5
**Confidence:** 4

**Summary:**

The paper presents a novel initialization scheme for parameterized quantum circuits optimized with variation quantum algorithms (VQA). The method employs a strategy based on Gaussian Mixture Models (GMMs) to initialize the parameters of an $L$ block $N$ qubit ansatz. The paper claims that for the considered ansatz, the initialization scheme avoids the barren plateau phenomenon (BP).

Theoretically, the authors prove lower bounds on the expectation of the gradient norm under three different assumptions for the observable in the loss function. By showing the expected gradient norm is nonzero, they are able to theoretically guarantee absence of the barren plateau at initialization

The authors also validate their initialization strategy on synthetic experiments. These experiments confirm that this initialization scheme indeed avoids the BP in practice as well.

**Strengths:**

1. As far as I am concerned, this is a novel contribution in the field of quantum machine learning.
2. Given it concerns only parameter initialization, the proposed initialization scheme is simple and easy to implement and compute efficient.
3. The method is backed up by solid theoretical guarantees which are also validated empirically through experiments. Also, the theoretical guarantees hold for rather practical situations, not just an idealized case.
4. The proofs in the appendix seem correct and are easy to follow.

**Weaknesses:**

1. Although your method is applied to a popular ansatz, it does not seem to generalize to other ansatz structures. Could you discuss potential for generalizing to different ansatz?
2. I found the paper to be written in a style that hinders understanding. There are various errors/inconsistencies in the notation and long inline math sections (around lines 158, 236, 250 for instance) that I personally found difficult to read. Please see the minor comments section for examples. It would be constructive to break up long inline math sections and give more context around complex mathematical equations.
3. It seems as though you did not validate your experiments using multiple runs/seeds. Number of runs/standard error/variance is not reported. If you indeed ran your experiments only once, this would be a major weakness of your experimental section; given your method is based on pseudorandom initialization, this would hinder the statistical validity of the results. Would it be possible for you to provide results from multiple runs along with error bars and confidence intervals?

Also here are some minor comments you may want to address for the final version:
* Generally, it would be clearer if you define all variables present in a theorem in the theorem statement for clarity
* In lines 86-100, you introduce the VQA problem and define the cost function. This should go in the notation/background section.
* Line 54 typo BP is underlined for citation
* Line 73 typo "expressibilityRagone et al."
* Line 227 typo, you have a citation in your big O
* Line 1131 "Theorem 1" should be "Lemma 1" I believe
* Generally inconsistent use of $cos$ and $\cos$
* In the proofs, inconsistent use of $I_S$ vs $I_s$

**Questions:**

* You claim your method "avoids barren plateau". However, the theoretical results only guarantee barren plateau is avoided at initialization. Do you have any insight on how this method may help avoid this phenomenon during training?
* Where does the eq for $f(\theta_{k+1})$ on line 87 come from?
* What is the importance of the result given by eq. (5)?
* How do you achieve a bound that does not depend on the number of qubits $N$ for Theorem 1? This seems surprising to me.
* You interchangeably use $O$ and $\mathbf{O}$ to describe observables. You also index this $O$ sometimes. Is this notation you did not define or simply a typo?
* In Figure 4, your method seems to reach the desired solution, but then as iterations continue, it diverges away before coming back. What explains this phenomenon you think?

---

### Official Review · Reviewer_EBag · 2024-11-03

**Soundness:** 1
**Presentation:** 1
**Contribution:** 2
**Rating:** 3
**Confidence:** 3

**Summary:**

This paper proposes using mixture of gaussians as initialization for variational quantum algorithms using mixture of gaussians. Theoretical results show the expectation of the norm of the gradient following the assumed distribution is lower bounded.

**Strengths:**

This paper proposes and proves that the mixture of gaussians as an initialization scheme avoids barren plateau (at initialization) even when the cost function is global. Experiments, even though settings are a bit unclear, seem to support their claim.

**Weaknesses:**

- My biggest concern is that the relation to the previous work [1] is hardly discussed. The authors should at least mention that [1] proposed for the first time the gaussian initialization precisely to prevent barren plateau at initialization—the exact setting this paper is addresses. I understand there are some differences discussed briefly starting from line 258, but even there the authors do not mention [1] uses Gaussian initialization. Such writing gives me the impression that the authors intentionally hide due to significant similarity with [1].
- I believe some people refer to “barren plateau” not only at the initialization but also more generally, i.e., vanish exponentially with the size of the system, c.f. [2, 3]. The authors should clearly state the “barren plateau” that they mention is only with regards to initialization; this is only mentioned in line 52 as: “the phenomenon of the barren plateau is characterized by the *randomized initialization* of parameters $\theta$ in VQAs,”…
- Continuing the above point, in my opinion it is misleading for instance to write in line 87 as:
$\theta_{k+1} = \theta_k - \alpha \nabla_\alpha f(\theta_k)$ … Therefore, typically $|| \nabla_\theta f (\theta_k) ||^2$ is used to determine whether the cost function can be updated.”
Given that this paper is only about initialization, what it really shows is that $|| \nabla_\theta f (\theta_0) \|^2||$ has significant magnitude, but the result does not say *anything* about $||\nabla_\theta f (\theta_k) \|^2||$ for $k > 1$.
- Please use parenthesized citations correctly; it’s very hard to read especially since the citation text colors are the same as the main text color
- Typos:
    - line 214: “Then We expand…”
    - line 231: wrong quotation marks for “inactive parameters”, etc

[1] Zhang, et al. (2022) “Escaping from the Barren Plateau via Gaussian Initializations in Deep Variational Quantum Circuits”

[2] Fontana et al. (2024) “Characterizing barren plateaus in quantum ansätze with the adjoint representation”

[3] Loracca et al. (2024) “A Review of Barren Plateaus in Variational Quantum Computing”

**Questions:**

- If I understand the result correctly, this only prevents “barren plateau” at initialization. I wonder if there is any comment the authors can make about the optimization trajectory (can you say anything about the norm of the gradient other than initialization?)
- I understand the above point is empirically argued e.g., in Figure 4. But the interpretation starting from line 413: “Moreover, the gradient norm remains within a relatively large range throughout the entire training process. This enables our approach to escape … vanishing gradient problem … . *These observations are entirely consistent with the conclusions drawn in Theorem 1*.” But isn’t Theorem 1 ONLY about initialization?
- In Theorem 1, the assumption of the parameters $\theta$ is that it follows $\mathcal{G}_1(\sigma^2)$ ? But this is just the Gaussian distribution $\mathcal{N}(0, \sigma^2)$. Could you explain how to interpret this? Why is this not compared to [1]? (other than one line sentence in line 228, “This is in stark contrast to the exponential lower bound $O(1/L^N)$ found in previous works for global cost functions Zhang et al. (2022a); Wang et al. (2023).)
- How is the experiment set up? Are these results of actually applying VQA to a quantum computer? Or are these some numerical simulations?
- Line 370: “…we compare our proposed method with… , Gaussian distribution $\mathcal{N} (0, \frac{1}{4S(L+2)})$ : is this variance taken from [1, Theorem 4.1]? It should really be cited clearly…
- Can you say anything about the solution quality? (converged $\theta$ after some iterations).

---

### Official Review · Reviewer_nSCi · 2024-11-05

**Soundness:** 2
**Presentation:** 2
**Contribution:** 2
**Rating:** 5
**Confidence:** 3

**Summary:**

This paper considers the variational quantum algorithms and deal with the barren plateau phenomenon. The new parameter initialization strategy is proposed combing with gaussian mixture models. The prove is provided that the initialization could avoids the barren plateaus problem.

**Strengths:**

The novel parameter initialization strategy is provided for the barren plateau phenomenon. And the prove is given,

**Weaknesses:**

The provided parameter initialization strategy is not clear in the figure 1. What’s more, the comparison with other methods is not given. Furthermore, the induced Gaussian mixture models is not the firstly introduced.

**Questions:**

1 In Figure2, how to determine the inactive parameters
2 in theorem1, what’s the impact of the partial of f(\theta)
3 in theorem 2, there is no definition of M, how to determine the value of M for the different number of layers.
4 what’s more, there is no comparison with other approaches,

---

### Note · Authors · 2024-11-13

I have read and agree with the venue's withdrawal policy on behalf of myself and my co-authors.